# Hit-and-run epigenetic editing prevents senescence entry in primary breast cells from healthy donors

Emily A. Saunderson [1], Peter Stepper[2], Jennifer J. Gomm[1], Lily Hoa [1], Adrienne Morgan[1], Michael D. Allen [1], J. Louise Jones[1], John G. Gribben[1], Tomasz P. Jurkowski [2] & Gabriella Ficz [1]

Aberrant promoter DNA hypermethylation is a hallmark of cancer; however, whether this is sufficient to drive cellular transformation is not clear. To investigate this question, we use a CRISPR-dCas9 epigenetic editing tool, where an inactive form of Cas9 is fused to DNA methyltransferase effectors. Using this system, here we show simultaneous de novo DNA methylation of genes commonly methylated in cancer, *CDKN2A*, *RASSF1*, *HIC1* and *PTEN* in primary breast cells isolated from healthy human breast tissue. We find that promoter methylation is maintained in this system, even in the absence of the fusion construct, and this prevents cells from engaging senescence arrest. Our data show that the key driver of this phenotype is repression of *CDKN2A* transcript *p16* where myoepithelial cells harbour cancer-like gene expression but do not exhibit anchorage-independent growth. This work demonstrates that hit-and-run epigenetic events can prevent senescence entry, which may facilitate tumour initiation.

[1] Barts Cancer Institute, John Vane Science Centre, Charterhouse Square, Queen Mary University of London, London EC1M 6BQ, UK. [2] Institute for Biochemistry and Technical Biochemistry, Department of Biochemistry, Faculty of Chemistry, University of Stuttgart, D-70569 Stuttgart, Germany. Correspondence and requests for materials should be addressed to T.P.J. (email: tomasz.jurkowski@ibc.uni-stuttgart.de) or to G.F. (email: g.ficz@qmul.ac.uk)

The epigenomic landscape is significantly perturbed during cancer development. In the case of DNA methylation, the best characterised epigenetic modification to date, the pattern of aberrant modifications is similar across different cancers[1]. In general, cancer cells have a hypomethylated genome, with some promoter CpG islands (CGIs) becoming hypermethylated[2–5] and the mechanism of this process is largely unknown. Since more than half of the coding genes contain a promoter CGI, which when methylated can inhibit their gene expression, hypermethylation can frequently result in tumour suppressor gene inactivation[6]. Previously, it has been difficult to dissociate passenger aberrant epigenetic changes from drivers in cancer initiation due to the lack of suitable experimental tools[7, 8].

Recent advances in epigenome editing are now enabling us to identify the role of DNA methylation in early tumorigenesis. The catalytic domain of de novo methyltransferase DNMT3A (in combination with DNMT3L in some studies) has been coupled to zinc finger proteins[9–12], TALEs (transcription activator-like effectors)[13], and most recently the catalytically inactive dCas9-CRISPR (clustered regularly interspaced short palindromic repeats) system[14–17], to introduce DNA methylation to a target locus. These studies have shown that DNA methylation can be successfully targeted, dependent on the combination of effector domains and localised chromatin confirmation, and that this has a direct effect on cell biology.

Successful DNA methylation editing using CRISPR has been shown in multiple cell lines[14–16, 18], primary T cells[16] and most recently in the mouse brain[18], although the maintenance of methylation is often limited without constitutive expression of the Cas9 construct[14, 15, 19]. Using CRISPR to co-target three effector domains, DNMT3A, DNMT3L and KRAB resulted in permanent hypermethylation after transient transfection in cell lines[16], whereas targeting only DNMT3A and KRAB did not, highlighting the importance of the local chromatin microenvironment in the effectiveness of these tools. Targeting DNA methylation with CRISPR has an interesting spreading effect as demonstrated recently, where a single gRNA resulted in DNA hypermethylation across the CGI[17]. These pioneering studies show the versatility and enormous potential for utilising CRISPR for epigenomic editing and have paved the way for our work interrogating the direct effect of DNA methylation on biological processes.

Here we transiently transfect dCas9 DNMT3A-3L (dCas9 3A3L) and show that DNA methylation can be targeted to multiple genes in primary breast cells isolated from healthy human tissue, resulting in long term hypermethylation and gene silencing. Cells are prevented from entering senescence and hyper-proliferate, a phenotype driven by p16 repression. Edited myoepithelial cells harbour cancer-like gene expression changes but are not immortal, indicating activation of early abnormal cellular processes which may enable cells to move towards transformation.

## Results

**Hypermethylation of tumour suppressors in primary cells**. To investigate whether promoter DNA hypermethylation can drive cellular transformation we established DNA methylation targeting in normal primary human myoepithelial cells isolated from healthy donors. The cell of origin in breast cancer is controversial but mammary stem cells may reside in the myoepithelial niche, contributing to both myoepithelial and luminal cell populations[20, 21]. We first optimised the transfection protocol in a myoepithelial cell line, 1089, cells which were isolated from healthy breast tissue and then immortalised[22, 23]. The dCas9 3A3L fusion plasmid contains the catalytic domain of mouse Dnmt3a and C-terminal domain of Dnmt3l (3A3L) coupled to a catalytically dead Cas9[17]. Cells were transiently transfected with the constructs and 5 days later analysed for DNA methylation changes (Supplementary Fig. 1a). Five guide RNAs (gRNAs) targeting the CGI overlapping the HIC1 gene promoter were designed to ensure DNA methylation spreading[14, 15] (Supplementary Fig. 1b) and this region was normally hypomethylated in parental 1089 cells (Supplementary Fig. 1b). dCas9 3A3L or the control 3A3LΔ (Supplementary Fig. 1c, 3A3LΔ construct inactive for methyltransferase function) were co-transfected with the gRNAs and DNA methylation was successfully targeted to the HIC1 promoter in 1089 cells (Supplementary Fig. 1d).

Transiently targeting DNA methylation can result in a spike of hypermethylation within 6–10 days before a rapid decline[14, 16, 19]. We checked this in 1089 cells and found HIC1 promoter hypermethylation at 5 days post-transfection when 25% of cells remained dCas9-positive (dCas9+ve; Supplementary Fig. 1e). By day 10 post-transfection DNA methylation was maintained although less than 1% of these cells were dCas9+ve, indicating that methylation had persisted in the absence of the fusion construct. However, at 15 days post transfection HIC1 was once again hypomethylated (Supplementary Fig. 1e). Our results validated previous findings that DNA methylation can be successfully targeted in cell lines; that in 1089 cells the modification was propagated for a few cell divisions but was lost in the absence of the effector fusion protein[17].

Unlike 1089 cells, primary myoepithelial cells go through a finite number of cell divisions before entering senescence (likely to be p16-induced based on work in primary breast cells[24]) and are challenging to transfect. We therefore allowed the primary cells to proliferate for 3 days before transfection (Fig. 1a). We designed five gRNAs to the HIC1 and RASSF1A promoters (Fig. 1b, c) and at 5 days post-transfection we identified DNA hypermethylation at both HIC1 and RASSF1A promoters when targeting dCas9 3A3L compared to dCas9 3A3LΔ (Fig. 1b, c; lower panels), indicating that transient transfection could be used to successfully target multiple genes in primary cells.

**Off-target DNA methylation effects of dCas9 3A3L targeting**. After demonstrating that we could successfully hypermethylate promoter CGIs in primary cells, we selected a panel of genes with substantial evidence suggesting a role in cellular transformation in breast cancer. Previous studies have identified DNA promoter hypermethylation at CDKN2A[25–28], PTEN[29–31], HIC1[32] and particularly RASSF1[3, 25, 26, 29, 33, 34] genes in early and late stage breast cancer. Furthermore, data mining[35] from 743 invasive breast carcinomas available from The Cancer Genome Atlas (TCGA) showed these genes become hypermethylated in some tumours (Supplementary Fig. 2). We limited our panel to four genes as this required transfection of 26 plasmids to cover the CGIs. Separately, we confirmed using 1089 cells that lipophilic based transfections resulted in the presence of dCas9 3A3L and pMACS (for magnetic sorting) proteins in the same cells (Supplementary Fig. 2e). We were therefore confident that most transfected cells contained all the constructs. In addition to HIC1 (Fig. 1b and Fig. 2a) and RASSF1A (Figs. 1c and 2c) we designed gRNAs against the CGI at the PTEN gene promoter (Fig. 2b); the CGI at RASSF1B and RASSF1C transcriptional start sites (TSS) (Fig. 2c); two CGIs overlapping CDKN2A at the TSS for p16INK4a (p16) and p14ARF (p14) transcripts; and a third partially hypomethylated downstream CGI (Fig. 2d). Gene expression analysis showed low levels of HIC1, RASSF1A and p14, compared to higher total RASSF1, PTEN and p16 expression in early passage cells (Supplementary Fig. 3a). We co-transfected all gRNAs and confirmed successful DNA methylation targeting at the HIC1 gene promoter after 5 days (Supplementary Fig. 3b). At

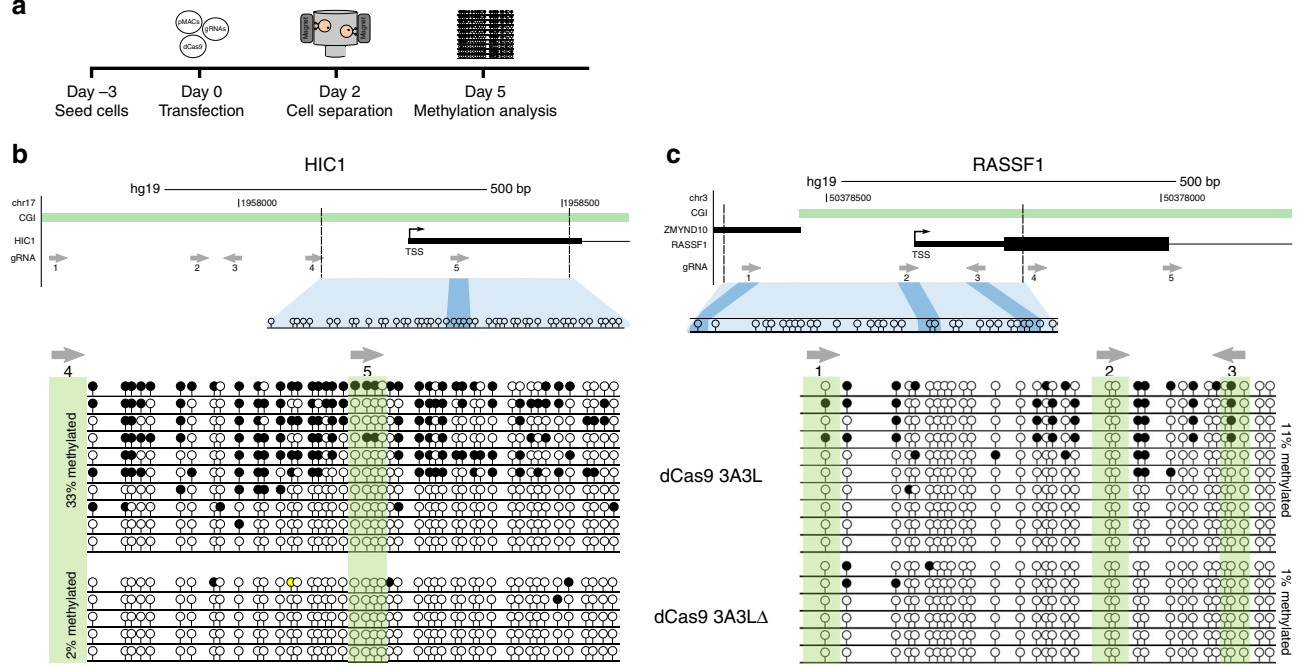

**Fig. 1** dCas9 3A3L targeted hypermethylation of *HIC1* and *RASSF1* in primary myoepithelial cells. **a** Schematic of experimental design, 10 plasmids encoding gRNAs targeting *HIC1* and *RASSF1A* promoters were transfected along with dCas9 3A3L or 3A3LΔ and the MACS (magnetic activated cell sorting) plasmid (pMACS) for magnetic selection. **b** A schematic depicting the gRNA location (grey arrows) in relation to the CGI (in green) and the TSS in the promoter of the *HIC1* gene; the arrows point towards the PAM sequence. The dotted lines represent the region analysed by bisulfite cloning and the locations of individual CpGs are scaled according to their genomic distribution. The dark blue shading depicts which CpGs are overlapped by gRNA #5. Lower panel shows bisulfite cloning results from the *HIC1* locus 5 days after transfecting primary myoepithelial cells from donor 1 with 10 gRNAs (targeting *HIC1* and *RASSF1*) and dCas9 3A3L or 3A3LΔ as indicated. **c** A schematic depicting gRNA in relation to CGI and TSS in the promoter of *RASSF1A*. Lower panel shows bisulfite cloning from *RASSF1A* 5 days after transfection of primary human myoepithelial cells with 10 gRNAs (targeting *HIC1* and *RASSF1*) and dCas9 3A3L or 3A3LΔ as indicated. Arrows and green shading represent overlap between gRNAs and CpGs. Each line represents a single clone, filled and empty lollipops represent a single methylated or unmethylated CpG, respectively

10 days post-transfection we analysed DNA methylation using the Infinium HumanMethylationEPIC (EPIC) array. Figure 2a–e show successful and significant hypermethylation of the target CGIs. Consistent with the cell line data and previous reports, we observed DNA methylation spreading from the site of the gRNA binding (e.g. upstream to the *PTEN* gRNA #1 in Fig. 2b). EPIC array probes associated with the target genes that had a significant and more than 20% methylation increase in dCas9 3A3L compared to the control dCas9 3A3LΔ cells are highlighted in Fig. 2e. The scatter plot shows a high correlation ($R = 0.99$) between the targeted and control cells, indicating little epigenome rearrangement at this stage (Fig. 2e).

Notably, we found 946 other CpGs that were consistently ≥ 20% hypermethylated in the dCas9 3A3L vs. dCas9 3A3LΔ transfected cells (0.1% of total CpGs measured), and only 185 CpGs were hypomethylated (Supplementary Data 1). The hypermethylated probes were significantly enriched in CGIs, CGI shores, and exons (above expected vs. all the EPIC probes) indicating a preference for hypermethylation at these genomic features (Fig. 2f). Out of all 946 off-targets only two genes contained more than 3 hypermethylated probes within 1–2 kilobases: the promoter of *G0S2* and the gene body of *C10orf41/ ZNF503-AS2* (Supplementary Fig. 4a, b). Hypermethylation events, other than our targets, may be genuine off-targets (some gRNA dependent or independent) or a consequence of the biological effect caused by on-target hypermethylation (as it has been previously reported for *G0S2* hypermethylation[36]). Identifying the extent of off-target effects associated with dCas9 based epigenetic editing is an important area for future work.

We checked whether DNA methylation targeting had suppressed *HIC1*, *RASSF1*, *PTEN* and *CDKN2A* associated gene expression at day 10 post-transfection using RNA-sequencing (RNA-seq) and targeted qPCR (Fig. 2g, h). *p16* expression increased significantly in 3A3LΔ control cells, indicating that primary myoepithelial cells indeed go into p16-induced senescence in culture. This expression was strongly suppressed in the 3A3L targeted cells (Fig. 2g). We found that *p14* was also downregulated in the 3A3L targeted cells. *RASSF1A* was repressed in both 3A3L and 3A3LΔ targeted cells compared to early passage (indicating downregulation in senescence, see Discussion), with a stronger repression in 3A3L targeted cells. The expression of other *RASSF1* transcripts was also significantly downregulated in 3A3L targeted cells compared to 3A3LΔ. *HIC1* and *PTEN* expression were similar between 3A3L targeted cells compared to 3A3LΔ. Taken together, the data suggest that transiently targeting dCas9 3A3L resulted in DNA methylation deposition to all 4 targeted genes, but only expression of *CDKN2A* and *RASSF1* were significantly reduced at 10 days post-transfection.

**Increased proliferation of dCas9 3A3L targeted cells**. We assessed whether our treatment had affected primary myoepithelial cell biology. Since a key hallmark of abnormality in cancer is uncontrolled cellular replication we quantitatively measured proliferation using a colorimetric based assay. We found that DNA methylation targeting resulted in rapid proliferation of cells after 15 days (Fig. 3a); the result was similar when using myoepithelial cells from two additional healthy donors (Fig. 3b). To compare the rate of population growth in

3A3L targeted cells compared to controls we measured population doublings (Fig. 3c). The growth of 3A3LΔ and untransfected cells slowed by 15 days post-transfection when cells start to senesce, staining positive for beta-galactosidase (β-gal; Fig. 3d) while cells targeted with dCas9 3A3L continue to proliferate for at least 35 days (up to 80 days). Regarding cell morphology, the

3A3L targeted cells are smaller and similar to that of early passage cells (Supplementary Fig. 5a, b) whereas the 3A3LΔ and untransfected cells become large and flat, typical of senescent cells (Fig. 3d). We observed a seeding density effect on this proliferative phenotype, with cells initially seeded at low density (780 cells cm$^{-1}$ ± 10%) proliferating significantly more than at high

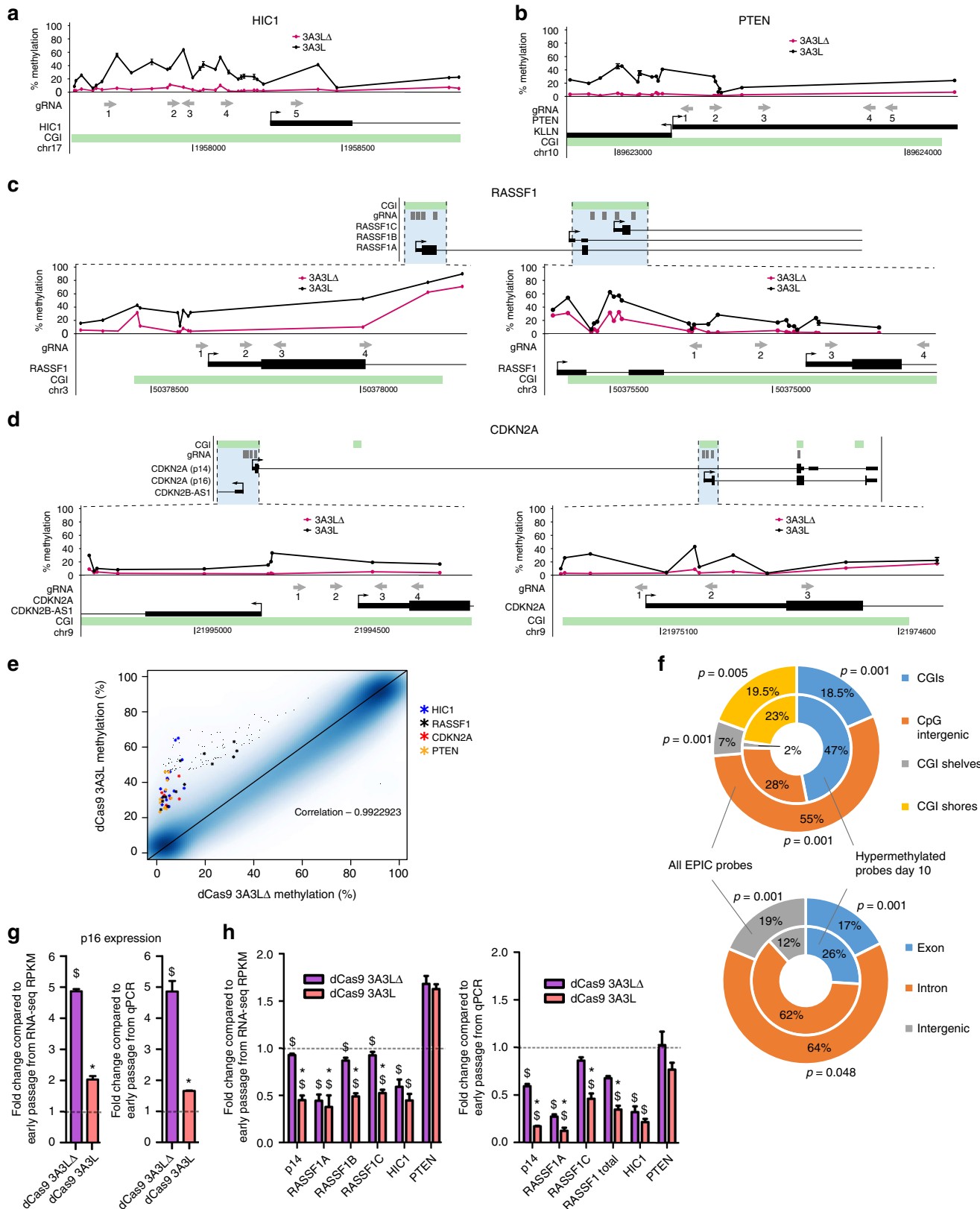

density (3120 cells cm$^{-1}$ ± 10%) after 15 to 20 days post-transfection (Supplementary Fig. 6a). We used the IncuCyte ZOOM™ live-cell imaging platform (Supplementary Movies 1 and 2) to track cell growth from 10 days post-transfection. This revealed that cells seeded at high density rapidly migrated together forming small clusters and did not proliferate (Supplementary Fig. 6b, c). The low density cells proliferated exponentially before reaching a plateau (Supplementary Fig. 6c, d). We speculate that the proliferation inhibition of myoepithelial cells seeded at high density is induced by paracrine signalling.

We performed additional experiments to test the contribution of unspecific effects to the proliferative phenotype. For these experiments (Figs. 3e and 6) we performed the transfections without pMACS magnetic enrichment to reduce cell loss and maximise the number of transfection conditions possible per experiment. Primary cells transfected with dCas9 3A3L without gRNAs did not proliferate, confirming that gRNA-independent binding of dCas9 3A3L does not prevent senescence (Fig. 3e). We targeted the panel of genes with a dCas9 3A3L catalytic mutant, where the C706A substitution inactivates the methyltransferase function and the targeted cells did not start proliferating (Fig. 3e; Supplementary Fig. 7). Finally, we used dCas9 without the 3A3L fusion construct and the 26 gRNAs and similarly saw no proliferation, excluding a dCas9-interference effect on gene expression (Fig. 3e). Interestingly, using dCas9 3A did result in senescence prevention indicating that DNMT3A targeting is sufficient in our system to drive the proliferative phenotype[17]. In summary, primary myoepithelial cells from three healthy donors consistently evaded senescence and were highly proliferative after DNA methylation targeting to the panel of tumour suppressor genes.

**Second cell stasis due to critically shortened telomeres.** Prolonged culturing of DNA methylation targeted cells resulted in reduced proliferation and cell cycle arrest after 60–80 days, with signs of cell crisis including: increase in cellular size, cytoplasmic vacuoles and multiple nuclei (Supplementary Fig. 8a). This growth profile is similar to a well described phenomena that occurs in a rare population of p16-hypermethylated human mammary derived epithelial cells (variant HMECs; vHMECs) originally isolated from healthy donor tissue[24, 37–43]. vHMECs arise spontaneously in culture; however, throughout our experiments we did not observe any spontaneous senescence escape in dCas9 3A3LΔ targeted or untransfected cells, even after maintaining deeply senescent cells for 63 days (Supplementary Fig. 8b). This may be due to genetic or epigenetic variations between donors, or differences in cell isolation and culture conditions. We found increased Ki-67$^{+ve}$ cells at 36 days post-transfection, a reflection of the rapid proliferation at this time point; but this was lost by 98 days, when cells are in permanent

cell cycle arrest (Supplementary Fig. 8c). We identified small clusters of CK8$^{+ve}$ cells 36 days post-transfection in 3A3L targeted cells, a marker normally associated with luminal cells, but these cells did not become more abundant at later time points. The outgrowing cells were permanently CK14$^{+ve}$, suggesting that cells remained myoepithelial in lineage.

To determine whether the outgrowing cells were transformed we assessed anchorage-independent growth. Cells were seeded at three time points: 2, 10 and 30 days post-transfection. In all cases there was no colony formation after 14 days (Supplementary Fig. 9a); however, 3A3L targeted cells that were rescued from the assay resumed proliferation in 2D culture vessels (Supplementary Fig. 9b), indicating that cells are dormant in 3D culture but retain proliferative potential.

In vHMECs the final cell cycle arrest is driven by critically shortened telomeres and immortalisation has been achieved by overexpressing hTERT[37, 44]. To assess whether this is the case in our cells we stably integrated a construct overexpressing hTERT with a green fluorescent protein (GFP) marker into 3A3L targeted cells from donor 1, and confirmed hTERT expression along with continued repression of p16 and p14, as well as for a separate experiment (Supplementary Fig. 10a–c). Over serial passages we found that overexpression of hTERT provided a growth advantage to the cells indicated by an increasing percentage of GFP$^{+ve}$ cells vs. control (Fig. 3f). The population doublings of hTERT overexpressing cells also increased vs. controls (Supplementary Fig. 10d). In summary, primary myoepithelial cells which evaded senescence entered a second, permanent, cell cycle arrest after several months which was independent of p16 expression and was alleviated by overexpressing hTERT.

**Targeted DNA methylation is propagated in dCas9 3A3L cells.** Since senescence prevention was consistently seen in primary myoepithelial cells we asked how permanent DNA methylation changes were in cells from independent donors during the exponential growth phase. We used the EPIC array to compare global DNA methylation in early passage primary myoepithelial cells from 2 donors and then at 37–38 days after DNA methylation targeting to the four genes. The early passage cells showed a similar DNA methylation profile (Fig. 4a, correlation 0.98), however by day 37–38 the DNA methylation profiles were more variable (Fig. 4b, correlation 0.90), but showed maintenance of methylation at RASSF1 and CDKN2A in both donors and less consistently at HIC1 and PTEN. Clustering all EPIC array data using principle component analysis (PCA) showed that the dCas9 3A3L and 3A3LΔ targeted cells clustered separately and that late passage cells have greater variability in the global DNA methylation pattern (Supplementary Fig. 11). We further validated the maintenance of DNA methylation after 37–40 days post-transfection at the four genes using targeted bisulfite

**Fig. 2** On- and off-target hypermethylation of tumour suppressor genes in primary cells. **a–d** EPIC array at targeted promoter regions of **a** HIC1, **b** PTEN, **c** RASSF1 and **d** CDKN2A. gRNA (grey arrows point towards PAM sequence) in relation to CGI (in green) and TSS of genes, percentage methylation from dCas9 3A3LΔ targeted cells (pink line) compared to 3A3L targeted cells (black line), 10 days post-transfection of primary myoepithelial cells from donor 1 and after magnetic sorting on day 2 (mean ± SEM; n = 3; excluding error bar if smaller than plotted point). **e** Scatter plot of EPIC data comparing dCas9 3A3LΔ to dCas9 3A3L targeted cells 10 days post-transfection. Significantly hypermethylated probes from HIC1, PTEN, RASSF1 and CDKN2A genes are highlighted on the plot ($\geq$ 20% increase; n = 3; Bumphunter; p < 0.01; Benjamini-Hochberg (BH) correction). **f** Overlap of the 946 EPIC array probes significantly hypermethylated in dCas9 3A3L targeted cells vs. 3A3LΔ (inner ring) compared to the percentage overlap of all probes in the EPIC array (outer ring) with CGIs, CGI shelves, CGI shores or CpG intergenic (top doughnut) and exons, introns or intergenic regions (bottom doughnut). Significant differences were calculated using regioner in R. **g, h** Gene expression of **g** p16 and **h** other target genes in primary myoepithelial cells from donor 1, 10 days after transfection with dCas9 3A3LΔ (purple) or dCas9 3A3L (orange). qPCR data is normalised to ACTB expression and shown as fold change compared to average gene expression in early passage cells (dotted line; mean ± SEM, n = 3; T-tests. *p < 0.01 dCas9 3A3L compared to 3A3LΔ; $^{\$}$p < 0.01 dCas9 3A3L or 3A3LΔ compared to early passage cells). RNA-seq data is shown as fold change of target gene transcripts from 3A3L or 3A3LΔ compared to average RPKM from early passage cells (dotted line; mean ± SEM, n = 3). Significance is from DESeq2 analysis with BH correction: *p < 0.001 3A3L compared to 3A3LΔ targeted cells; $^{\$}$p < 0.001 3A3L or 3A3LΔ compared to early passage cells

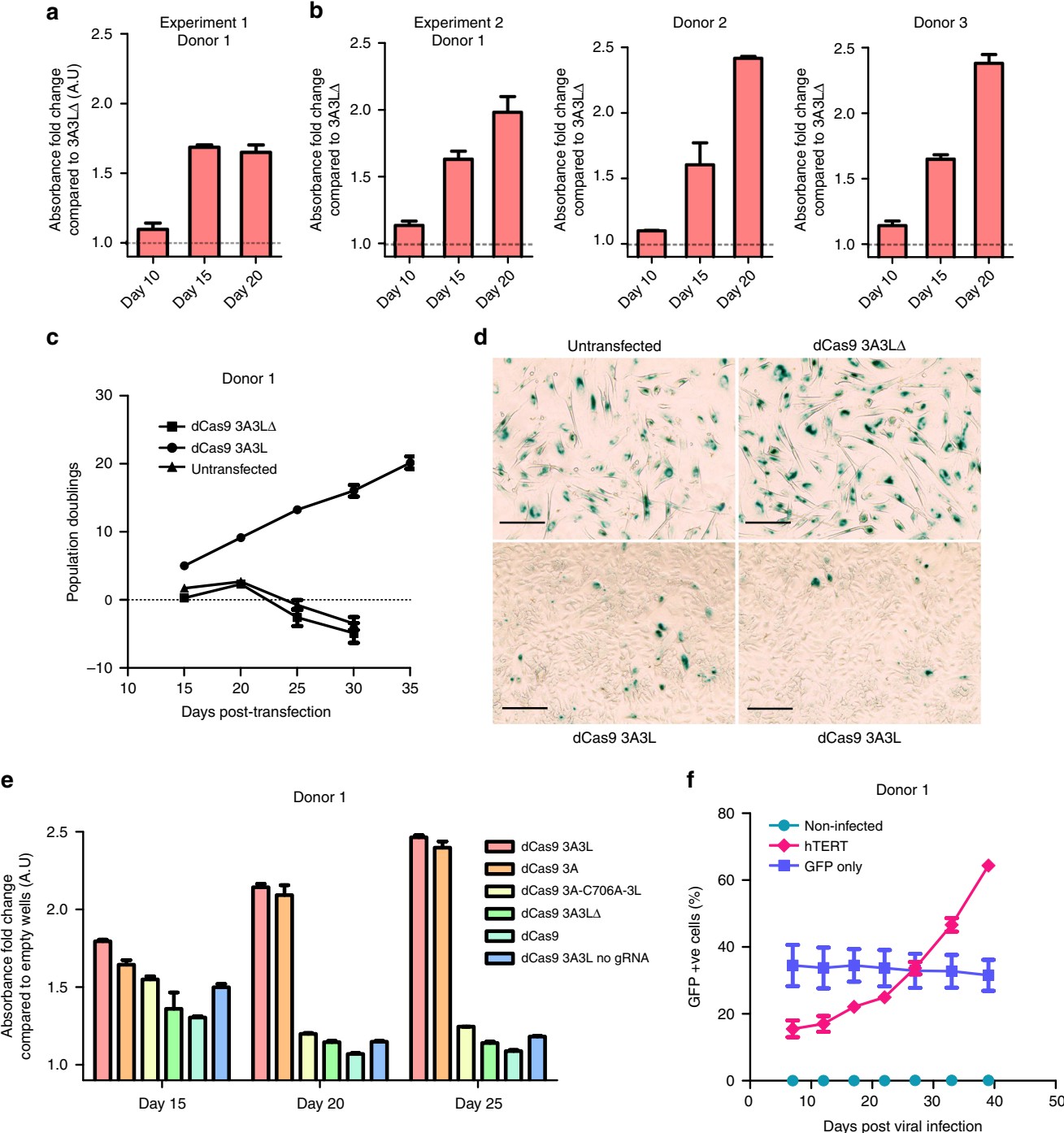

**Fig. 3** dCas9 3A3L or dCas9 3A targeting prevents senescence and increases proliferation. **a**, **b** Proliferation was assessed using a colorimetric assay 10, 15 and 20 days after targeting dCas9 3A3L to *HIC1*, *RASSF1*, *PTEN* and *CDKN2A* and magnetic sorting in primary myoepithelial cells from donors 1, 2 and 3. Dotted line represents the average absorbance from three dCas9 3A3LΔ targeted wells at each timepoint. The graphs show the average difference in absorbance as fold change compared to 3A3LΔ average (mean ± SEM, *n* = 3). **c** Cumulative population doublings over time for untransfected primary myoepithelial cells and those transfected with dCas9 3A3L or 3A3LΔ targeting the four genes without magnetic sorting from donor one. Fifty thousand cells were seeded at the start of each passage and cells counted after 5 days (mean ± SEM, *n* = 3, where error bars are smaller than points plotted they are not shown). (**d**) Light microscopy images of β-gal staining from untransfected (top left) primary myoepithelial cells from donor 1 and those transfected with dCas9 3A3L (bottom two) or 3A3LΔ (top right) targeting the four genes without magnetic sorting 20 days after transfection, blue/green colouring shows β-gal+ve cells; scale bar represents 125 μm. **e** Proliferation was assessed using a colorimetric assay 15, 20 and 25 days after targeting the four genes with dCas9 3A3L (red), dCas9 3A (orange), dCas9 3A-C706A-3L (yellow), dCas9 3A3LΔ (green) or dCas9m4 (pale green), or targeting only dCas9 3A3L without gRNAs (blue) without magnetic sorting using donor 1 myoepithelial cells. The data is shown as fold change compared to the average absorbance from 3 wells without cells (mean ± SEM, *n* = 3). **f** The percentage of GFP+ve cells after infection with no virus (green), hTERT with GFP marker (pink) or GFP control (purple). At infection day 0 cells are 38 days post-transfection with dCas9 3A3L targeting the four genes (mean ± SEM; *n* = 3, error bars are not shown where smaller than points plotted)

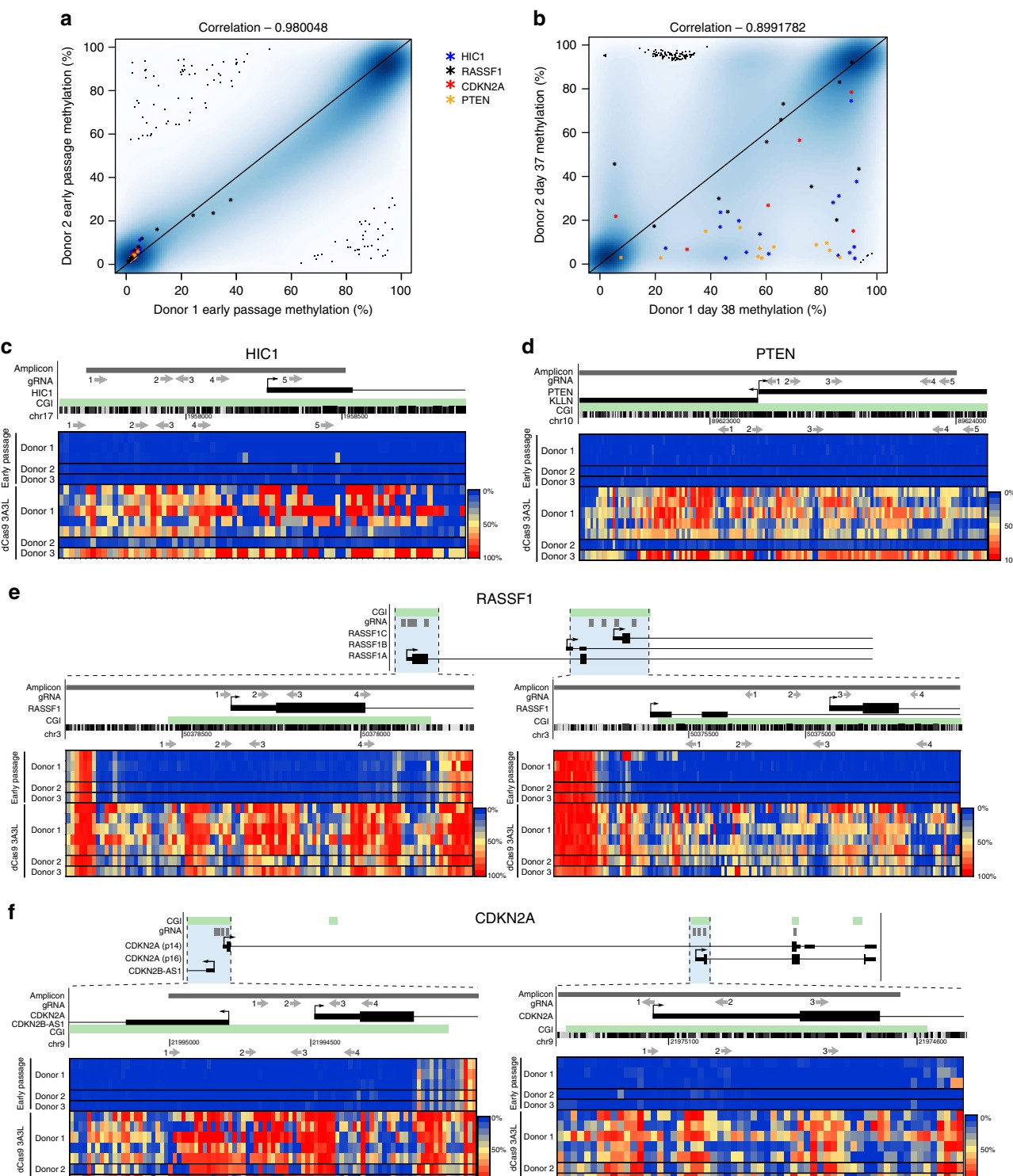

**Fig. 4** DNA methylation is consistently maintained at *HIC1*, *RASSF1* and *CDKN2A*. **a** Scatter plot of DNA methylation from all EPIC array probes comparing donor 1 and donor 2 early passage primary myoepithelial cells (*n* = 1). The probes from the target genes highlighted on the plot are those that were found to be significantly hypermethylated at day 10 after dCas9 3A3L vs. 3A3LΔ targeting as shown in Fig. 2e. **b** Scatter plot of DNA methylation from all EPIC array probes comparing donor 1 and 2 primary myoepithelial cells 38 and 37 days respectively after targeting dCas9 3A3L to *HIC1*, *PTEN*, *RASSF1* and *CDKN2A*. The same probes from **a** are highlighted (*n* = 1). **c**–**f** Targeted bisulfite sequencing at **c** *HIC1*, **d** *PTEN*, **e** *RASSF1* and **f** *CDKN2A*. The localisation of the bisulfite amplicon (dark grey bar) and gRNA (grey arrows pointing towards the PAM sequence) in relation to CGI (green bar) and TSS of genes is shown. Each rectangle represents the methylation % indicated by the colour key of a single CpG and each line is data from a single replicate. The top 5 lines show data from early passage (passage 2) primary myoepithelial cells from donors 1, 2 and 3, the bottom 7 lines show data from primary cells 37–41 days after transfection with dCas9 3A3L targeting *HIC1*, *RASSF1*, *PTEN* and *CDKN2A*. Top lines of data from donor 1 early passage and dCas9 3A3L targeted is from the same samples as the EPIC array data in Fig. 4a, b. Bisulfite sequencing data from donor 2 is from the same samples as the EPIC array data shown in Fig. 4a, b

sequencing and showed DNA methylation remained at these regions in multiple replicate experiments using donor 1 myoepithelial cells, as well as in myoepithelial cells from donors 2 and 3 (Fig. 4c–f). This also confirmed our gene targets were hypomethylated in untransfected primary myoepithelial cells from the three donors. Nevertheless, we noticed some variability in DNA methylation maintenance between donors such as *HIC1* and *PTEN* in donor 2 compared to donor 1 and 3 (Fig. 4b, c, d). We also found consistent hypermethylation at the *C10orf41/ZNF503-AS2* gene body in all donor myoepithelial cells transfected with dCas9 3A3L and validated a hypomethylated region in both

control and targeted cells (Supplementary Fig. 12a and b). We could not find consistent hypomethylation in edited cell replicates (Supplementary Fig. 12c, d). Gene expression analysis of the target genes in early passage vs. dCas9 3A3L targeted (days 35–37 post-transfection) showed significant gene repression of all targeted genes (*CDKN2A*, *RASSF1*, *PTEN* and *HIC1*) in donors 1 and 3 (Supplementary Fig. 13a, b). Interestingly, for donor 2, DNA methylation at *PTEN* and *HIC1* was not well maintained at late passages (Fig. 4c, d), consistent with their re-expression in targeted cells (Supplementary Fig. 13c). Furthermore, in donor 2, *CDKN2A* associated transcripts and *RASSF1A* were significantly

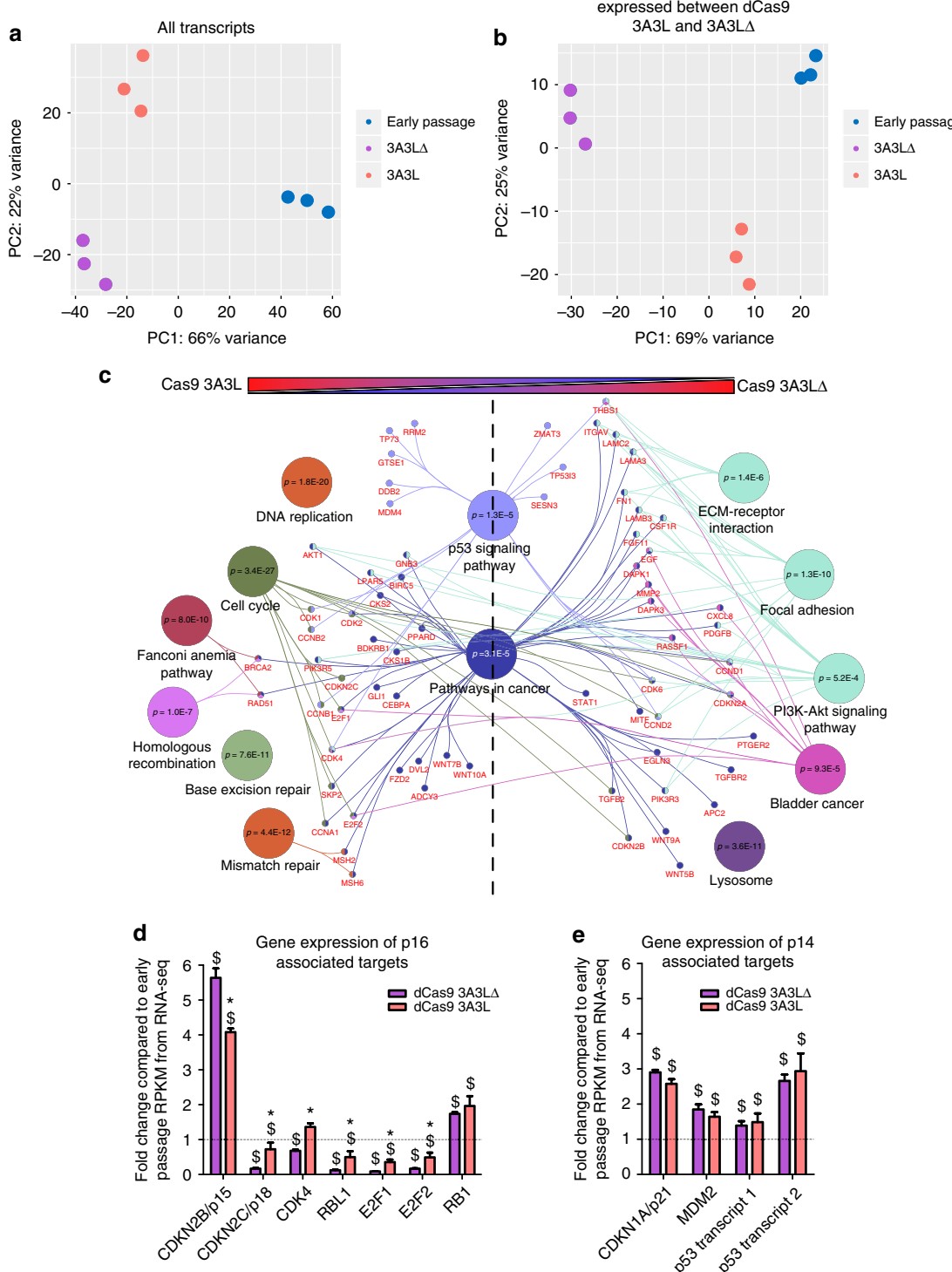

downregulated but other *RASSF1* transcripts were not repressed despite maintained DNA methylation at the promoter (Supplementary Fig. 13c).

In summary, targeting DNA methylation via dCas9 3A3L to the panel of genes was permanently maintained in primary myoepithelial cells from 3 donors, albeit with some inconsistences between donors, and this resulted in repression of all gene targets in donors 1 and 3.

**dCas9 3A3L cells activate cancer-like gene expression**. To interrogate the intracellular signalling changes that occured when cells were prevented from entering senescence, we performed RNA-seq using primary myoepithelial cells from donor 1, 10 days post-transfection including pMACS magnetic sorting on day 2. We included a third group of untransfected myoepithelial cells (early passage cells) as a control (Supplementary Fig. 14). We used the data to cluster samples by PCA and found that 3A3L and 3A3LΔ groups clustered together at principle component 1 (PC1; Fig. 5a) indicating that 60% of the variance could be explained by the time in culture. It is important to mention that as indicated by p16 expression in Fig. 2g, some of the control 3A3LΔ cells are already senescent at 10 days post-transfection. Gene ontology (GO) analysis revealed both 3A3L and 3A3LΔ targeted cells had upregulation of abnormal and cancer-associated signalling processes compared to early passage cells (Supplementary Fig. 15a; Supplementary Data 2); with substantial overlap between pathways differentially regulated in 3A3L and 3A3LΔ targeted cells compared to the early passage group (Supplementary Fig. 15b).

Interrogating the differential gene expression between 3A3L and 3A3LΔ targeted cells revealed a subset of transcripts (1925 genes; Supplementary Data 3) which we used to perform a second PCA. Analysing this subset of transcripts showed that the 3A3L targeted cells clustered more closely to the early passage cells at PC1, accounting for 69% of the variance (Fig. 5b); this observation was corroborated by hierarchical clustering (Supplementary Fig. 16a). Expression of this subset of genes in 3A3L more closely matches the profile in early passage cells (Supplementary Fig. 16b). This suggests that 3A3L targeting to *HIC1*, *RASSF1*, *PTEN* and *CDKN2A* suppresses senescence and prevents gene expression changes associated with senescence. We used GO analysis to identify the signalling pathways associated with the genes differentially expressed between 3A3L and 3A3LΔ targeted cells. This identified upregulation of cell cycle- and DNA damage-linked transcripts in DNA methylation targeted cells (Fig. 5c). 3A3LΔ targeted cells had increased transcription associated with aberrant processes and cancer, indicating that the cells entering senescence had abnormal intracellular signalling compared even to 3A3L targeted cells, although as mentioned above both 3A3L and 3A3LΔ groups had abnormal signalling compared to early passage (Supplementary Fig. 15).

To further elucidate the driving events behind the observed phenotype in DNA methylation targeted cells, we used the RNA-seq data to analyse expression of transcripts downstream of, or associated with p16, p14 and RASSF1A. Here we found that the majority of these transcripts were differentially expressed when comparing transfected cells to early passage (Fig. 5d, e). Many associated and downstream factors from p16 signalling were also differentially expressed between 3A3L and 3A3LΔ targeted cells (Fig. 5d; $p = 5.7 \times 10^{-8}$ (Fisher's exact test) compared to 7 genes picked at random from total transcripts). *CDKN2B* (*p15*), another cell cycle regulator, was significantly upregulated as 3A3LΔ targeted cells approached senescence; but repressed in 3A3L targeted cells. This finding has parallels with another study, whereby *p15* was upregulated in deeply senescent HMEC cells compared to early passage, and expression of *p15* was significantly repressed following knockdown of p16 by siRNA[45], suggesting negative feedback regulation of p15 after p16 repression. Surprisingly, factors downstream from p14 such as *MDM2* and *p53* (*TP53*) were upregulated in both 3A3L and 3A3LΔ targeted cells compared to the early passage group (Fig. 5e); implying that p53 signalling is activated regardless of p14 status in the cell. Similarly, transcripts associated with *RASSF1A* signalling were not differentially expressed between 3A3L and 3A3LΔ targeted cells (Supplementary Fig. 17).

In summary, this data indicates that dCas9 3A3L and 3A3LΔ targeted myoepithelial cells had similar overall transcriptional profiles by 10 days post-transfection, and this was associated with abnormal cellular signalling. However, targeting DNA methylation to the four genes prevented cells from entering senescence by maintaining a subset of transcript expression at levels similar to early passage cells.

**p16 epigenetic silencing prevents senescence entry**. From our RNA-seq data and substantial evidence in the literature[37, 42, 43, 46, 47], it seemed likely that the prevention of cellular senescence we observed was driven by *p16* promoter hypermethylation and subsequent repression. To investigate this, we targeted DNA methylation to *CDKN2A*, *RASSF1*, *PTEN* and *HIC1* genes individually and assessed proliferation. As predicted, targeting all four genes or targeting the CGIs associated with *CDKN2A* prevented donor 1 and 2 myoepithelial cells from entering senescence; whereas targeting DNA methylation to *RASSF1*, *PTEN* or *HIC1* alone did not (Fig. 6a).

Next, we asked whether *p16* hypermethylation was the sole event required to drive senescence prevention in different donors. We individually targeted each *CDKN2A* CGI with gRNAs (as shown in Fig. 6b). When targeting *p14* alone, only donor 1 myoepithelial cells were consistently prevented from entering senescence (Fig. 6c) and targeting *CDKN2A* CGI3 had no effect (Supplementary Fig. 18a, b). Targeting *p16* alone with 3 gRNAs was sufficient to prevent senescence and enable proliferation (Fig. 6d) comparable to cells transfected with dual *p14*-*p16* specific gRNAs (Fig. 6e); however, as shown in Fig. 6f and

**Fig. 5** RNA-seq gene expression analysis of dCas9 3A3L treated and control myoepithelial cells. **a** PCA analysis of RNA-seq data using all transcripts from early passage (blue), dCas9 3A3LΔ targeted (purple) and dCas9 3A3L targeted (orange) primary myoepithelial cells from donor 1, 10 days post-transfection targeting *HIC1*, *RASSF1*, *PTEN* and *CDKN2A* (including magnetic sorting at day 2). Each circle represents an individual biological replicate (n = 3). **b** PCA analysis using only data from transcripts which are differentially expressed between dCas9 3A3L and 3A3LΔ targeted cells as identified by *DESeq2* (p < 0.001; with BH correction); data are shown as early passage (blue), dCas9 3A3LΔ targeted (purple) and dCas9 3A3L targeted (orange) cells. Each circle represents an individual biological replicate (n = 3). **c** GO analysis of transcripts which are differentially expressed between dCas9 3A3L and 3A3LΔ targeted cells. GO terms associated with transcripts highly expressed in dCas9 3A3L targeted cells (left panel) and 3A3LΔ targeted cells (right panel) are shown. P value was calculated using Cytoscape (v. 3.4.0) and the add-in package Cluego (v. 2.2.5), which utilises the KEGG pathway database. **d**, **e** Gene expression of transcripts associated with **d** p16 signalling and **e** p14 signalling from the RNA-seq data 10 days post-transfection in primary myoepithelial cells from donor 1. Data is shown as fold change of gene expression from 3A3LΔ (purple) and 3A3L (orange) compared to average RPKM data from early passage cells (dotted line; mean ± SEM, n = 3). Significance is from *DESeq2* analysis: *p < 0.001 3A3L compared to 3A3LΔ targeted cells; $^{\$}$p < 0.001 3A3L or 3A3LΔ compared to early passage

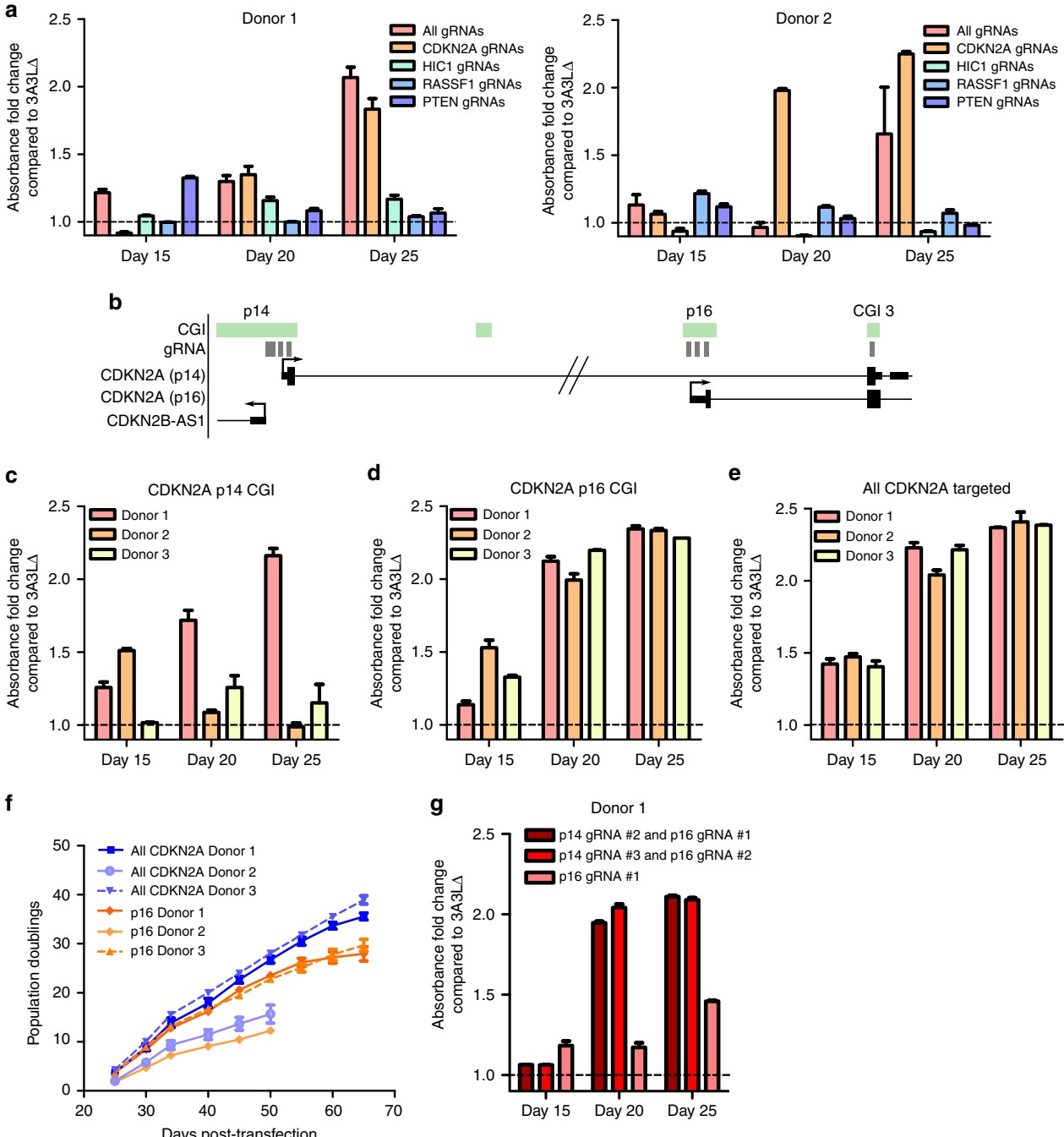

**Fig. 6** Hypermethylation and repression of p16 drives the proliferative phenotype. **a** Proliferation was measured using a colorimetric assay 15, 20 and 25 days after targeting dCas9 3A3L or 3A3LΔ to *HIC1*, *RASSF1*, *PTEN* and *CDKN2A* (red), *CDKN2A* only (orange), *HIC1* only (green), *RASSF1* only (blue) or *PTEN* only (indigo) in donor 1 (left graph) or donor 2 (right graph) primary myoepithelial cells. Dotted lines represent the average absorbance from three dCas9 3A3LΔ targeted wells at each timepoint. The graphs show the average difference in absorbance as fold change of 3A3L compared to 3A3LΔ average (mean ± SEM, n = 3). **b** Schematic showing the structure of the *CDKN2A* locus and the location of the gRNAs (grey boxes) in relation to CGIs (green bar) and TSS of alternative transcripts p14 and p16. **c-e** Proliferation was assessed 15, 20 and 25 days after targeting dCas9 3A3L or 3A3LΔ to **c** p14, **d** p16 or **e** the whole CDKN2A locus in donors 1, 2 and 3 primary myoepithelial cells. Dotted lines represent average from three dCas9 3A3LΔ targeted wells. The fold change of 3A3L compared to 3A3LΔ average is plotted (mean ± SEM, n = 3). **f** Cumulative population doublings for cells from donors 1, 2 and 3 transfected with dCas9 3A3L targeting p16 with gRNAs #1, #2 and #3 (orange lines) or targeting the entire CDKN2A locus (8 gRNAs; blue lines). Fifty thousand cells were seeded at the start of each passage and cells counted after 5 days (mean ± SEM, n = 3, error bars excluded where smaller than points). **g** Proliferation was measured 15, 20 and 25 days after targeting dCas9 3A3L or 3A3LΔ to CDKN2A using p14 gRNA #2/p16 gRNA #1 (magenta), p14 gRNA #3/p16 gRNA #2 (red) or p16 gRNA #1 (pink) in donor 1 myoepithelial cells. Dotted line represents the average from three dCas9 3A3LΔ targeted wells. Data is plotted as fold change of 3A3L absorbance compared to 3A3LΔ average (mean ± SEM, n = 3)

Supplementary Fig. 18c, dual targeting of *p14-p16* resulted in significantly more proliferation in donor 1 and 3 respectively compared to targeting only *p16* (Supplementary Fig. 18c; *p* < 0.0001; two-way ANOVA with Bonferroni post-hoc test). Gene expression analysis 30 days post-transfection confirmed that *p16* was repressed after methylation targeting to *p14-p16* or *p16*, whereas *p14* was repressed in *p14-p16*-targeted cells (Supplementary Fig. 19a–c). Expression of the functionally related *p15* transcript increased in some replicates when methylation was targeted to *p14-p16* compared to *p16* alone (Supplementary Fig. 19d), demonstrating variation in intracellular processes after *p14-p16* vs. *p16* methylation targeting.

To exclude a contribution of dCas9 3A3L-mediated off-target methylation to the phenotype, we used two sets of gRNAs with different sequences and targeted *CDKN2A* with dCas9 3A3L (Fig. 6g). Both combinations resulted in the primary cells proliferating, excluding a role for off-target methylation effects in the proliferative phenotype, as each gRNA set has different off-targets. Notably, when we used a single gRNA targeting *p16* (gRNA #1) these cells evaded senescence (Fig. 6g). This suggests that *p16* hypermethylation is sufficient to prevent senescence entry in myoepithelial cells and that targeting both *p14* and *p16* promoter CGIs may increase the rate of proliferation.

Finally, to extend our finding of the role of *CDKN2A* hypermethylation beyond myoepithelial proliferation we targeted hypermethylation in primary luminal cells (which line the duct within the breast) from two donors. Using luminal cells we found that targeting *CDKN2A* with dCas9 3A3L resulted in proliferation whereas targeting dCas9 3A3LΔ did not (Supplementary Fig. 20a). Primary luminal cells were consistently able to proliferate beyond the expected capacity *in vitro* (Supplementary Fig. 20b), but also reached a final cell cycle arrest. Interestingly, 3A3L transfected luminal cells were able to grow in colonies in the anchorage independence assay (Supplementary Fig. 20c), indicating that the luminal cells may have undergone physiological changes more akin to transformation. Our results suggest that targeting *p14-p16* hypermethylation and prevention of senescence entry is applicable to other primary cell types although the physiological response might be cell dependent.

## Discussion

The expanding field of epigenome editing allows us to ask functional questions that have hitherto been technically challenging. In this study, we show that hit-and-run DNA methylation targeting in normal primary myoepithelial cells from multiple donors causes heritable promoter hypermethylation at *CDKN2A*, *RASSF1* and *HIC1*, with permanent repression of *CDKN2A* and *RASSF1* associated transcripts. This prevents a subset of gene expression changes associated with senescence and blocks senescence entry. Finally, we demonstrate that hypermethylation and repression of *p16* drives this phenotype and repressing the whole *CDKN2A* locus increases the proliferative capacity of primary cells (Fig. 7).

Our ability to successfully target epigenetic modifications is no longer in question[9, 13–18, 48] but evidence for DNA methylation maintenance without constitutive expression of the targeting machinery has been limited[13–16, 18, 19]. In our study, repressing *CDKN2A* in primary myoepithelial and luminal cells enabled crucial anti-proliferation barriers to be overcome. Utilising the biological context of the cell in this way separates our finding from recent work demonstrating heritable DNA methylation using three effector constructs[16]. Targeted DNA methylation was consistently maintained at the *RASSF1* locus, but this was unable to prevent senescence entry; however repression of *RASSF1A* may influence other biological processes aside from proliferation. Our

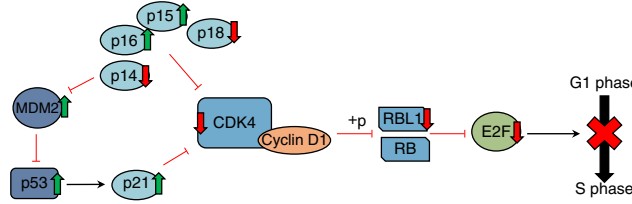

10 days after 3A3LΔ transfection — senescence

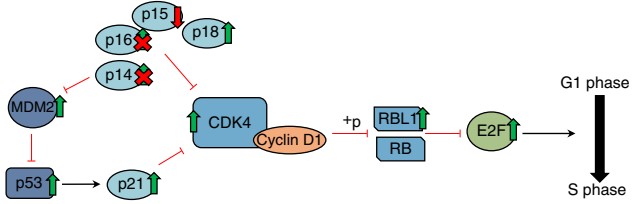

10 days after 3A3L transfection — senescence prevention

**Fig. 7** Model of epigenetic editing mediated senescence prevention. A schematic showing the main transcriptional changes that occur 10 days after DNA methylation is targeted to *CDKN2A* by dCas9 3A3L which prevents senescence entry

observation that *RASSF1A* was also repressed as cells enter senescence may support the theory that senescence-associated changes are not necessarily cancer-preventative and could explain why *RASSF1A* hypermethylation is found at early stages of breast neoplasms[3, 26, 29]. Although the *PTEN* and *HIC1* promoters became hypermethylated at 10 days post-transfection in donor 1, gene expression was not affected in the bulk population. However, we did find significant repression after 35–37 days post-transfection in donors 1 and 3 (Supplementary Fig. 13a, b); this nevertheless raises the possibility that some genes can be more resistant to DNA methylation repression.

From our EPIC array data, we found almost 1000 CpGs with ≥ 20% hypermethylation in dCas9 3A3L vs 3A3LΔ targeted cells 10 days post-transfection (0.1% of total CpG probes). These may not be random events as there is a 70% overlap between 3A3L replicates when the top 300 hypermethylated CpGs are analysed (Supplementary Fig. 21, probes unmethylated in control). Currently, it is not possible for us to distinguish whether these are the result of transient dCas9 3A3L off-target binding or whether these are consistent biology-driven hypermethylation events occurring after cell cycle re-entry driven by *CDKN2A* repression. Based on our data showing senescence prevention using different combinations of *p14-p16* gRNAs, and a single *p16* gRNA (Fig. 6g) we believe that we can exclude possible dCas9 3A3L-mediated off-target methylation from contributing to the proliferative phenotype.

Our experiments show that the outgrowing primary myoepithelial cells are not immortal and go into permanent cell cycle arrest. A similar growth pattern has been studied extensively in vHMECS[37, 49]; previous work showed that vHMEC cells (which have escaped senescence via spontaneous p16 promoter hypermethylation) also eventually go into permanent cell cycle arrest and have critically shortened telomeres and gross chromosomal aberrations[40]. Our DNA methylation targeted cells in cell cycle arrest show a similar phenotype based on increased cellular size, cytoplasmic vacuoles and multiple nuclei and partial proliferation rescue after the overexpression of hTERT; supporting the hypothesis that senescence escape is the first step in malignancy *in vivo*, promoting genetic abnormalities that can destabilise the genome and transform cells. Accordingly, several studies have shown that *p16* hypermethylation is required before

overexpression of c-MYC or hTERT can immortalise primary HMECs[36, 50].

Our work may provide foundations for understanding DNA methylation-driven early changes in breast cancer in vivo. It is well known that p16 is commonly misregulated in human cancer and p16 is considered a tumour suppressor, however, the emerging picture is more complex. Clearly, a lack of p16 promotes cancer as mice models have shown that knocking out p16 (and to a greater extent p14-p16) increases spontaneous cancer incidence[51]; however there are additional epigenetic or genetic hits required before disease emergence. In other studies, too much p16 is also detrimental and increases disease risk[52], including cancer[53, 54]. It may be that using the levels of p16 alone is not sufficient to draw decisive and consistent conclusions about cancer risk[55, 56]. Whether DNA methylation targeted to p14-p16 increases disease susceptibility is unknown, however it should be noted that p16 promoter hypermethylation is found in histologically normal breast tissue[27, 39].

Although our experiments indicate that DNA methylation targeting is the driving modification behind the phenotype, successful epigenetic editing is dependent on chromatin conformation[16, 48], therefore low or repressed p16 expression may be a prerequisite for DNA hypermethylation in primary cells, a hypothesis that is supported by work using HMECs[57]. Limitations on the number of primary cells available and the lack of reliability for single cell histone modification analysis makes it difficult to address this question. Interestingly, the proliferative phenotype we observe is not restricted to primary myoepithelial cells, it also occurs after targeting DNA methylation to CDKN2A in primary luminal cells, suggesting a more universally relevant biological mechanism; an important finding since it is not fully understand whether myoepithelial cells can contribute to luminal-type breast cancers[20, 21]. Exploring our observation that targeting DNA methylation induces a more cancer-like phenotype in the luminal cells will be an exciting direction for future work.

In summary, we have used dCas9 3A3L to successfully target promoter hypermethylation in primary breast cells demonstrating the usefulness of this construct as part of our rapidly expanding molecular toolkit to delineate cause from consequence in diseases such as cancer. We show that hit-and-run epigenetic editing can be used to initiate aberrant cellular processes and we predict that this will pave the way for identifying key epigenetic drivers in cancer, with the potential for development of new treatments and therapies.

## Methods

**Cell culture**. Primary human myoepithelial and luminal cells were isolated from reduction mammoplasty tissue as previously described[58]. Briefly, breast tissue was chopped into 1 cm pieces and incubated overnight with 1 mg ml$^{-1}$ collagenase and hyaluronidase in RPMI media (5% fetal bovine serum (FBS), 100 units ml$^{-1}$ penicillin, 0.1 mg ml$^{-1}$ streptomycin, 5ug ml$^{-1}$ amphotericin B) at 37°C. Following 3x washes in medium to remove enzymes and fat, $3 \times 30$ min sedimentation steps at $1g$ were used to separate the denser organoids from single cells such as blood cells and fibroblasts. Organoids were digested to a single cell suspension using 0.05% trypsin and 0.025% EDTA in PBS including 0.4 mg ml$^{-1}$ DNase for 10–20 min at 37 °C. To isolate a pure primary myoepithelial cell population, the single cell suspension was incubated with mouse anti-CD10 primary antibody (#mca1556, Bio-Rad), conjugated to sheep anti-mouse immunoglobulin G Dynabeads at a ratio of 2:1 (cells to beads) for 30 min at 4 °C to label primary myoepithelial cells. Magnetically labelled myoepithelial cells were collected using magnetic separation and the remaining cells were incubated with Epithelial Enrich Dynabeads for 30 min at 4 °C to label and isolate luminal cells. Healthy donors 1, 2 and 3 were aged 24, 21 and 20 respectively at time of surgery, and donor 4 (luminal cell experiment only) was aged 28. The donor cells used for each experiment is specified in the Figure or Figure legend. Primary myoepithelial cells were at passage 2 at the start of all experiments and were cultured on collagen coated plates in human mammary epithelial cell medium (Thermo Fisher Scientific) supplemented with bovine pituitary extract (50 μg ml$^{-1}$; Thermo Fisher Scientific), hydrocortisone (0.5 μg ml$^{-1}$), insulin (5 μg ml$^{-1}$), epidermal growth factor (EGF;

10 ng ml$^{-1}$), gentamicin (10 μg ml$^{-1}$) and amphotericin B (0.5 μg ml$^{-1}$). Immortalised myoepithelial cell line 1089 (created and provided by Professor Michael O'Hare[22]) was generated from primary human myoepithelial cells by mutant SV40 LT antigen (U19tsA58) and hTERT overexpression, before sorting using integrin β4–labelled sheep anti-mouse magnetic beads to purify the myoepithelial cell population[23]. Cells were maintained in Ham's F12 media containing L-glutamine, FBS (10% v/v), hydrocortisone (1 μg ml$^{-1}$), insulin (5 μg ml$^{-1}$) and EGF (10 ng ml$^{-1}$). Primary luminal cells were at passage 1 at the start of all experiments and were cultured on collagen coated plates in DMEM-F-12 medium supplemented with fetal bovine serum (10% v/v), hydrocortisone (0.5 μg ml$^{-1}$), insulin (5 μg ml$^{-1}$), EGF (10 ng ml$^{-1}$), apo-transferrin (10 μg ml$^{-1}$), amphotericin B (2.5 μg ml$^{-1}$) and penicillin-streptomycin (100 units ml$^{-1}$ penicillin, 0.1 mg ml$^{-1}$ streptomycin). Cells were maintained at 37 °C in a 5% CO$_2$ humidified incubator and were routinely testing for absence of mycoplasma. Informed consent was obtained from all human participants who donated their tissues to the BCI Breast Tissue Bank.

**Plasmid construction**. The dCas9 3A3L fusion and dCas9 3A-C706A-3L mutant were generated and used as previously described[17] (Supplementary Fig. 1c). Briefly, the Dnmt3a–Dnmt3L single-chain construct[10] was introduced to the M-SPn-Cas9-VP64 plasmid (Addgene plasmid #48674)[59] via PCR amplification from the ZNF-Dnmt3a3L plasmid[10] and cloning using the Gibson assembly. The dCas9 3A3LΔ plasmid contained a duplicated 100bp region within the Dnmt3l region, rendering the construct catalytically inactive due to 3A3L mis-folding (Supplementary Fig. 22). The gene promoter sequences containing CGIs from human HIC1, RASSF1, CDKN2A and PTEN were extracted using the UCSC genome browser[60]. gRNA sequences were designed using a CRISPR design tool (http://crispr.mit.edu)[61] and sequences were selected based on minimal predicted off-target binding, with particular attention given to avoiding gRNAs with off-targets predicted within CGIs. Sequences were selected to give good coverage of each CGI, with 100–200 bps between each gRNA (Supplementary Table 1). The gRNA plasmids were synthesized using complementary ssDNA oligonucleotides with overhangs which were annealed and cloned into the empty gRNA plasmid (Addgene plasmid #41824) using Gibson Assembly or T4 DNA ligase following the manufacturer's protocol, giving one plasmid per gRNA. Correct gRNA sequence in the final plasmid construct was confirmed by Sanger sequencing.

**Transfections**. For 1089 transfection, cells were seeded $8.5 \times 10^5$ cells in a T75 flask and transfected the following day using 12.7 ug total plasmid, (35% Cas9 3A3L or 3A3LΔ, 41% equimolar pooled HIC1 gRNAs (8.2% per gRNA) and 24% magnetic activated cell sorting (MACS) plasmid Kk.II (pMACS; MACSelect system; Miltenyi Biotech); jetPrime (Polyplus) ratio 1:2.6), magnetic sorting resulted in at least 80% transfected cells. For experiments using primary myoepithelial cells, 6 well plates were collagen coated before seeding $2 \times 10^4$ cells well$^{-1}$. After 3 days, primary cells were pre-treated for 4 h with hyaluronidase (3mg ml$^{-1}$) to increase transfection efficiency before cells were washed in phosphate buffered saline (PBS) and incubated with the transfection cocktail for 4 h. The proportions of dCas9 3A3L or 3A3LΔ, gRNAs and pMACS were adjusted depending on the number of gRNAs. We transfected plasmid quantities based on total pmol (0.45 pmol total per well; jetPrime ratio 1:9.67) due to the difference in size between the dCas9 plasmids and gRNAs. The plasmid cocktails were as follows: for 26 gRNAs: 0.037 pmol dCas9 3A3L or 3A3LΔ, 0.374 pmol gRNAs (0.0144 pmol per gRNA), and 0.038 pmol pMACS; for 5 gRNAs (targeting HIC1 or PTEN): 0.1 pmol dCas9 3A3L or 3A3LΔ, 0.25 pmol gRNAs (0.05 pmol per gRNA), and 0.1 pmol pMACS. For 8 gRNAs (targeting RASSF1 or all CDKN2A locus): 0.077 pmol dCas9 3A3L or 3A3LΔ, 0.296 pmol gRNAs (0.037 pmol per gRNA), and 0.077 pmol pMACS. For targeting the p14 CGI (4 gRNAs): 0.11 pmol dCas9 3A3L or 3A3LΔ, 0.225 pmol gRNAs (0.056 pmol per gRNA), and 0.11 pmol pMACS. For targeting the p16 CGI (3 gRNAs): 0.11 pmol dCas9 3A3L or 3A3LΔ, 0.226 pmol gRNAs (0.067 pmol per gRNA), and 0.11 pmol pMACS. For CDKN2A CGI3 targeting (1 gRNAs): 0.17 pmol dCas9 3A3L or 3A3LΔ, 0.11 pmol gRNA, 0.17 pmol pMACS. Magnetic enrichment was performed 2 days after transfection for some experiments and this is stated in the Figure legends, giving at least 80% transfection efficiency. For experiments which did not involve magnetic sorting the same total amount of pmols was transfected as above without the pMACS and the amount of other plasmids was increased proportionally, this gave 20–25% transfection efficiency. For the combination targeting shown in Fig. 6g we used p14 gRNA #2 and p16 gRNA #1 or p14 gRNA #3 and p16 gRNA #2 (2 gRNAs): 0.225 pmol dCas9 3A3L or 3A3LΔ, 0.225 pmol gRNAs (0.1125 pmol per gRNA). For p16 targeting using only a single gRNA shown in Fig. 6g, we used p16 gRNA #1: 0.26 pmol dCas9 3A3L or 3A3LΔ and 0.19 pmol gRNA #1. For the multiple Cas9 targeting experiment shown in Fig. 3e we used 0.413 pmol of gRNAs (all 26x gRNAs, 0.016 pmol per gRNA) and 0.037 pmol of dCas9 3A3L, 3A3LΔ, 3A, 3A-C706A-3L or the Cas9 quadruple mutant (dCas9m4) with 0.413 pmol of gRNA. We used 0.45 pmol of dCas9 3A3L for transfection without gRNAs. For experiments using primary luminal cells the transfection procedure was the same but with a total of 0.8 pmol of plasmid and 1:7 of pmol:jetPrime reagent as these cells are more challenging to transfect.

**Bisulfite cloning**. Genomic DNA was extracted (PureLink; Thermo Fisher Scientific) before bisulfite conversion (Imprint, Sigma-Aldrich) following manufacturer's guidelines. Regions within *HIC1* or *RASSF1* gene promoters were amplified using bisulfite-specific primers (Supplementary Table 2). Purified bisulfite-treated DNA was added to a mastermix (1x reaction buffer, 2 mM MgCl₂, 0.025 units μl⁻¹ HotStarTaq (Qiagen), 200 μM of each deoxynucleoside triphosphate (dNTP), 0.20 μM forward and reverse primers) and amplified using a thermocycler. PCR products were purified and sub-cloned using p-GEMT (Promega) and 5-alpha competent E. coli, before single positive colonies were picked and sequenced using Sanger sequencing and vector specific primers.

**Global DNA methylation analysis**. The Illumina Infinium HumanMethylationEPIC (EPIC) array was used to analyse global DNA methylation differences. Primary myoepithelial cells from donor 1 were seeded and transfected 3 days later with dCas9 3A3L or 3A3LΔ, all 26 gRNAs and pMACS as described. Each replicate (n = 3) was performed as an entirely separate transfection experiment. Successfully transfected cells were magnetically sorted 2 days post-transfection and replated. 10 days post-transfection cells were harvested and either genomic DNA or RNA (see below RNA-seq) was extracted. Since the EPIC array requires at least 250ng of genomic DNA and there are limitations on the number of primary cells for each experiment and low transfection efficiencies, genomic DNA for dCas9 3A3LΔ targeted replicates 2 and 3 were pooled (these are replicates 2 and 3 in the RNA-seq analysis i.e. not pooled for the RNA-seq experiment). Another separate transfection of dCas9 3A3LΔ and gRNAs was used as replicate 3 in the EPIC array. Data shown in Fig. 4a are from early passage myoepithelial cells (i.e. passage 2 cells grown for 3 days before harvesting) from donors 1 and 2 (n = 1). Data shown in Fig. 4b are from myoepithelial cells from donor 1 and 2 transfected with dCas9 3A3L, all 26 gRNAs and pMACS before magnetic enrichment and culturing for 38 days (donor 1) or 37 days (donor 2) (n = 1). DNA (400–500 ng) was bisulfite converted and DNA methylation was quantified using the EPIC BeadChIP (Illumina) run on an Illumina iScan System using the manufacturer's standard protocol. Raw IDAT files were processed using the *ChAMP* package (v. 2.8.2) in R using BMIQ normalisation[62] to generate methylation β-values and calculate significant DMRs. All downstream analysis was conducted using the hg19/GRCh37 human genome assembly. Annotation of DMRs and significant overlap at genomic features was calculated using *regioneR* (v. 1.8.0). PCA analysis was performed using DNA methylation data from all probes.

**Targeted bisulfite sequencing**. Passage 2 primary myoepithelial cells from donor 1, 2 and 3 were seeded and either cultured for 3 days before harvesting (early passage) or they were transfected after 3 days with dCas9 3A3L, all 26 gRNAs and pMACS. Transfected cells were magnetically sorted after 2 days and then cultured until 35–41 days post-transfection with regular passaging before harvesting cells and extracting genomic DNA. DNA was bisulfite converted and amplified by PCR using primers specific to target regions (Supplementary Table 2). PCR products were purified using SPRI beads (Agencourt AMPure XP, Beckman Coulter). Amplicons were end repaired and A-tailed (Agilent Technologies) before ligation to unique TruSeq HT double indexed adapters. Samples were pooled and PCR amplified with 8 cycles using Illumina specific primers and quantified with the NEBNext Library quantification kit for Illumina (New England Biolabs). Sequencing was performed on an Illumina MiSeq with 150bp paired-end reads at Barts and the London Genome Centre (London, UK). Reads were quality trimmed with Trim Galore (v. 0.4.1) and mapped to the human genome using Bismark (v.0.14.4) before visualization and quantification using SeqMonk (v. 0.34.1).

**Proliferation assay**. For measuring proliferation in experiments which involved magnetic sorting (Fig. 3a, b, Supplementary Fig. 6a) on day 2 after transfection and immediately after magnetic sorting, primary myoepithelial cells were seeded in triplicate at 780 cells cm⁻¹ (± 10%). Proliferation was assessed on days 10, 15 and 20 after transfection using different wells for each timepoint and using CellTiter-Blue (Promega) by diluting reagent 1:5 with media and incubating for 1–4 h. Proliferation was quantified by measuring the absorbance at 550nm and 620nm as a reference and data were normalised to the average absorbance of 3 wells without cells. For seeding density experiments (Supplementary Fig. 6), cells were magnetically sorted 2 days post-transfection and cultured until 10 days post-transfection before reseeded in a 96 well plate at 3120 cells cm⁻¹ (± 10%; high density), 1560 cells cm⁻¹ (± 10%; medium density), 780 cells cm⁻¹ (± 10%; low density). Proliferation was quantified at 15 and 20 days after transfection. For experiments which did not involve magnetic sorting (a change in method to reduce cell death and increase the number of conditions possible for each experiment), we transfected primary myoepithelial cells as described and then cultured cells until 10 days post-transfection before reseeding cells in triplicate at 780 cells cm⁻¹ (± 10%) in a 96 well plate. Proliferation was assessed at days 15, 20 and 25 after transfection using CellTiter-Blue as described. For luminal data, cells were seeded and after 3 days were transfected with dCas9 3A3L or 3A3LΔ and all 26 gRNAs as described. After 15 days cells were reseeded in a 96 well plate in triplicate and proliferation assessed at 20, 25 and 30 days after transfection.

**Population doublings**. Population doublings were calculated over multiple passages by seeding 50,000 cells at the start of each passage and counting the number of cells at the end of the passage (i.e. 5 days later for primary myoepithelial cell experiments). Doublings were calculated using the formula:

$$log_2 \frac{\text{no. of cells at end of passage}}{\text{no. of cells at start of passage}}$$

Data are plotted as the cumulative population doubling over time. Negative or decreasing cumulative population doublings occur when fewer cells were present at the end of the passage than at the start (i.e. when cells are not growing).

**β-galactosidase (β-gal) staining**. Primary myoepithelial cells were seeded and either cultured as an untransfected control or transfected with dCas9 3A3L or dCas9 3A3LΔ with 26 gRNAs. Cells were cultured for 20 days and then fixed in culture plates for overnight β-gal staining using a kit (New England Biolabs) following the manufacturer's instructions. Images of the staining were taken using a light microscope.

**hTERT overexpression**. The Platinum-A (Plat-A) amphotropic retroviral packaging cell line (Cell Biolabs) was maintained in DMEM containing 4.5 mg ml⁻¹ glucose, ʟ-glutamine, 10% v/v FBS, 10 μg ml⁻¹ blasticidin and 1 μg ml⁻¹ puromycin. A retroviral vector expressing hTERT with a GFP marker (Addgene #69809) and a GFP-only control vector (Addgene #21654) were transfected into the Plat-A cells using jetprime. Fresh media without selection antibiotics was put on Plat-A cells 24 h after transfection. Virus-containing supernatant was removed 24 h later, filtered and mixed in a 1:1 ratio with primary myoepithelial media before adding to primary cells. 7 days after infection and every 5 days subsequently, cells were passaged and the total number counted. Fifty-thousand cells were reseeded after each passage and the cumulative population calculated as described. The remaining cells at each passage were stained with 4′,6-diamidino-2-phenylindole (DAPI) to exclude dead cells and the percentage of GFP⁺ᵛᵉ cells was assessed for both hTERT and control infected cells using LSR Fortessa (BD Biosciences) following the manufacturer's instructions.

**RNA-sequencing (RNA-seq)**. Each replicate (n = 3) was performed using donor 1 myoepithelial cells but as an entirely separate transfection experiment performed on different days. For RNA-sequencing, primary myoepithelial cells at passage 2 were seeded and after 3 days, early passage cells were harvested. On the day of early passage harvesting, the remaining cells were transfected with dCas9 3A3L or dCas9 3A3LΔ, all 26 gRNAs and MACS plasmid as described. Cells were magnetically sorted 2 days post-transfection and then cultured until 10 days post-transfection at which point they were harvested. As described in the 'Global methylation analysis' section, dCas9 3A3L or 3A3LΔ targeted cells from this experiment were also used for DNA methylation analysis on the EPIC array. For RNA-seq, total RNA was extracted using Direct-zol (Zymo Research) and DNase treated (Thermo Fisher Scientific), before mRNA was isolated from 1.5 μg of total RNA using mRNA DIRECT (Thermo Fisher Scientific) and fragmented with RNA fragmentation reagent. First strand cDNA synthesis was performed with SuperScript III First-Strand Synthesis System and 3 μg μl⁻¹ random hexamers (Thermo Fisher Scientific) followed by second strand synthesis with DNA polymerase I and RNase H. After purification using SPRI beads, the double stranded cDNA was ligated to in house designed adapters (based on TruSeq Indexed adapters (Illumina)) using NEBNext Ultra II (NEB) followed by 15 cycles of amplification and library purification. Sequencing was performed on an Illumina HiSeq4000 with 75bp paired-end at the Oxford Genomics Centre (The Wellcome Trust Centre for Human Genetics, Oxford, UK). Genomic mapping of short reads was performed by the Oxford Genomics Centre using bwa (v. 0.7.0) to the human genome (GRCh37). RNA-sequencing analysis was performed using SeqMonk and the inbuilt RNA-seq quantitation pipeline which automatically corrects for the total number of reads. The R package *DESeq2* was used to calculate differentially expressed transcripts between groups (p < 0.001 with Benjamin Hochberg multiple test correction). PCA analysis was performed using either all transcripts or the subset of transcripts significantly differentially expressed between 3A3L and 3A3LΔ targeted cells. PCA plots were calculated by exporting the raw counts from SeqMonk before *rlog* transforming and using the PCA plotting function in *DESeq2*. For GO analysis, we used Cytoscape (v. 3.4.0) and the add-in package Cluego (v. 2.2.5), which utilises the KEGG pathway database. Hierarchical clustering and violin plots were created using SeqMonk.

**Gene expression analysis**. For target gene expression analysis using qPCR, RNA was extracted from samples as above and DNase treated before cDNA conversion. cDNA was mixed with a SYBR Green mastermix before qPCR to assess gene expression. Gene expression was normalised to a single housekeeping gene, either *ACTB* or *GAPDH*, to save material and primers were designed using Primer3plus (Supplementary Table 3). Data was analysed using the Pffaffl method[63].

**IncuCyte analysis of cell density effect on growth**. For IncuCyte analysis of growth with different seeding density (Supplementary Fig. 6b–d), cells were magnetically sorted 2 days post-transfection and cultured until 10 days post-transfection before reseeded in a 96 well plate at 3120 cells cm$^{-1}$ ($\pm$ 10%; high density), 1560 cells cm$^{-1}$ ($\pm$ 10%; medium density), 780 cells cm$^{-1}$ ($\pm$ 10%; low density). Cells were cultured in an incubator connected to an IncuCyte Live Cell Analysis System. At 10x magnification, 1 image hour$^{-1}$ of three locations per well were taken over 13 days and the confluency of the cells was calculated by creating an image collection and processing definition using the IncuCyte ZOOM confluence processing software.

**Anchorage independent growth**. Primary myoepithelial or luminal cells were seeded 600 cells per 24 well plate in a 300 μl mixture of 50% complete culture media and 50% Matrigel Matrix (Corning). After 1 h in the incubator 300 μl of media was carefully layered on top; media was changed weekly and images were taken 12–14 days post-seeding as described in Figure legend.

**Flow cytometry for dCas9 3A3L or 3A3LΔ**. The myoepithelial 1089 cell line was trypsinised and fixed in 4% paraformaldehyde for 20 min at 4°C. After washing in staining buffer (3% v/v FBS, 0.02% w/v sodium azide in PBS), cells were permeabilised in 88% v/v methanol at −20°C for 10 min before a further wash in staining buffer. Mouse anti-Cas9 antibody (1:1000; Active Motif #61577) was added and cells were incubated for 30 min at room temperature. After washing, cells were incubated with anti-mouse 2° Alexa 647 (1:500; Thermo Fisher Scientific #A21235) for 30 min at room temperature. Flow cytometry was performed using LSR Fortessa following the manufacturer's instructions.

**Immunostaining**. For immunocytochemistry, primary myoepithelial cells were grown in 24 well plates before fixing in 10% neutral buffered formalin for 10 min. Wells were pre-treated with hydrogen peroxide (100 μl H$_2$O$_2$ 30% w/v in 10ml PBS) to reduce endogenous peroxidase activity. Cells were permeabilised (0.1% Triton X-100, PBS, 15 min), before blocking (5% bovine serum albumin (BSA), 15% normal goat serum, PBS, 1 h). Cells were incubated with 1 μg ml$^{-1}$ 1° mouse antibodies (anti-CK14 (#PA0074 Leica Biosystems); anti-Ki-67 (clone MIB-1 Aligent Technologies); anti-CK8 (clone 2E4 Sigma-Aldrich) or anti-IgG (#A9044 Sigma-Aldrich)) in blocking buffer for 1 h. After washing, 2° antibody was added (1:1000; biotin conjugated goat anti-mouse (Sigma-Aldrich)) for 1 h before incubating with Vectastain ABC kit (1 h; Vector Laboratories) and developing with DAB (3,3'-diaminobenzidine tetrahydrochloride; (Vector Laboratories)). For immunofluorescence, 1089 cells were attached to slides using a cytospin, before fixing with 4% w/v paraformaldehyde in 1 mM PBS. Cells were permeabilised using 0.2% Triton X-100 before adding anti-Cas9 (1:200, mouse; Active Motif #61577) for 1 h. After adding 2° antibody Alexa 388 anti-mouse (1:200, Life Technologies #A11001), slides were dried and coated in vectorshield mounting media containing DAPI (Vector Laboratories) and visualised using Eclipse Ci (Nikon) fluorescent microscope.

**Statistical analysis**. Significance testing was performed using Prism (v.5.04) and Student's T-test, one-way ANOVA or two-way ANOVA with Bonferroni post-hoc tests as specified in the Figure legends. All experiments were repeated at least three times, unless stated otherwise. Where applicable, data are plotted as mean ± SEM.

**Data availability**. RNA-seq and EPIC array data that support the findings of this study have been deposited in Gene Expression Omnibus (GEO) the accession code GSE100209 (https://www.ncbi.nlm.nih.gov/geo/query/acc.cgi?acc=GSE100209). All other data that support the findings of this study are available from the corresponding authors upon reasonable request.

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

## Acknowledgements

This work was supported by MRC NIRG award to G.F. [Grant Ref: MR/M01892X/1] and Baden-Würtemberg Stiftung [95011370 to T.P.J.] and Barts Cancer Institute. The authors wish to acknowledge the role of the Breast Cancer Now Tissue Bank in preparing and making available the cells used in the generation of this publication. We thank Dr. Miguel Branco, Miss Hemalvi Patani and Dr. Michael Rushton for their advice on preparation of the manuscript.

## Author contributions

Study was conceived and designed by G.F. and E.A.S. Experimental work was done by E. A.S. Materials and epigenetic editing expertise were provided by T.P.J. Expertise in primary breast cell biology and breast cancer was provided by J.L.J. Contributions to experimental work were made by J.J.G., P.S., A.M., L.H. and M.D.A. Statistical analysis was performed by E.A.S. Research was supervised by G.F., J.G.G. and T.P.J. Manuscript was written by G.F. and E.A.S. All authors contributed to the editing of the manuscript and approved its final version.

## Additional information

**Competing interests:** The authors declare no competing financial interests.

