## [Peer Review File · Nature Communications]

Reviewers' comments:

Reviewer #1 (Remarks to the Author):

In this manuscript Saunderson et al report the successful use of the fusion protein dCas9 DNMT3A/3L to direct DNA methylation and repression of putative tumour suppressor genes in primary myoepithelial cells. They observe that DNA methylation targeted cells have increased proliferation compared to control cells, bypass senescence, and present a subset of genes with an expression profile similar to the one of early passage cells. Finally, the authors identify that p16 and p14 repression drive the senescence bypass.

It is well known that epigenetic dysregulation is a hallmark of cancer, but so far evidence that DNA methylation alterations can drive cancer has been correlative and indirect. For this reason, the work presented in the manuscript is of particular interest, as it shows, for the first time to the knowledge of the reviewer, that manipulation of DNA methylation at tumour suppressor gene promoters leads to an abnormal proliferative phenotype.

Although the data presented are fairly convincing, the manuscript would greatly benefit from further work to robustly support the claims made. In addition parts of the manuscript are confusing and it contains a large number of typos. It therefore, requires clarification and editing.

Major points:

1. The authors describe initially the use of dCas9 3A3L to target DNA methylation in SKOV3 and 1089 cell lines (Fig.1). The successful use of this tool in cell lines has been reported previously (Stepper et al PMID: 27899645) and their results in these cell lines differ from those in primary myoepithelial cells (it is transient and limited to single genes). This difference is not followed up in the rest of the manuscript. Therefore, this initial part of the manuscript appears redundant and of marginal interest.

2. The authors suggest the genes they target are not methylated in myoepithelial cells by examining of Roadmap Epigenomics data. However, this is a single replicate and they do not confirm this is the case for their myoepithelial cell cultures.

3. A major concern of epigenome-editing techniques is the contribution of off-target effects on the phenotype observed. From the MeDIP-seq results it is clear that dCas9 3A3L transfection into primary myoepithelial cells results in the methylation of many genomic regions other than the ones targeted by their cocktail of gRNAs. The authors suggest these off-target events are not caused by gRNA dependent dCas9 3A3L recruitment because they overlap minimally with computationally predicted off-target binding sites. Their evidence is dependent on the accuracy of the computational algorithm used (in this case Hsu et al 2013 from the Zhang lab). However, given that the prediction of CRISPR off-target binding events is in its infancy, the use of single, older algorithm only provides weak evidence towards this conclusion. The alternative, proposed by the authors is that dCas9 3A3L is able to methylate genomic regions independently of its gRNAs. The best control to rule out an involvement of these aspecific methylation events in senescence escape would be to transfect primary myoepithelial cells with dCas9 3A3L but no gRNAs, before analysing their proliferation. The experiments presented in Fig. 7a-b provide some evidence that this would not be the case. However, no methylation data are shown for these experiments. Comparison of methylation profiles resulting from dCas9 3A3L transfections with and without gRNAs would help clarify this issue. Similarly, ChIP for dCas9 3A3L with and without sgRNAs would provide further evidence to determine whether these off-target events are gRNA dependent or independent. Note, the nomenclature used by the authors here is slightly confusing as regions methylated by dCas9 3A3L independently of gRNAs are strictly still off-target events. In the manuscript the authors reserve off-target for predicted gRNA dependent sites. They should use gRNA dependent and independent off-target effects or similar terminology.

4. From the data presented in Supp Fig 7, the magnitude of methylation change at the PTEN promoter looks similar to HIC1, RASSF1 and CDKN2A. However, the authors suggest it does not gain methylation. We note that the authors empirically set the p-value threshold for differential methylation based on one locus, HIC1. How robust is the discrimination of differential methylation to false positives, can they validate with further loci using an independent assay?

5. The authors notice that dCas9 3A3L target cells have increased proliferation rate compared to control cells dCas9 3A3L Δ , but that this phenotype is dependent on cell seeding density. However, the effect of the seeding density observed does not fit with the model proposed in the manuscript. According to the model, targeting of both p16 and p14 enables senescence bypass senescence. Targeted methylation is however only partially efficient and it is reasonable to suppose that hypermethylation of both p14 and p16 will occur in a small fraction of cells. The number of double-targeted cells would be expected to increase with the number of plated cells, ie the number of escapees and therefore overall proliferation rate should increase with seeding density. However, the authors describe the opposite effect, proliferation and, presumably, the number of escapees are higher with low seeding density.

As the seeding density is an important parameter in their experiments, it should be specified for the further experiments performed (eg are MeDIP and RNAseq data derived from cells plated at low or high densities?). It would also aid future research in this area if cell-seeding densities used were specified (eg what are low, medium and high in Fig 4)?

6. The authors conclude that targeting of p14 + p16 by dCas9 3A3L enables escape from senescence in primary myoepithelial cells. However, they do not show that untransfected or dCas9 3A3L-delta cells undergo senescence. The reference given in the introduction for untransfected cells (Brenner et al 1998) discusses HMECs rather than primary myoepithelial cells isolated with the protocol used here. Can the authors provide data using markers to properly demonstrate that these cells do indeed undergo senescence?

7. As discussed by the authors, cultured HMEC cells enter senescence but a subpopulation can escape (vHMECs), before entering a final period of growth arrest which has been termed agonescence. The authors observe similarities between the growth phenotype of the dCas9 3A3L targeted myoepithelial cells and vHMECs. This would be expected as HMECs are believed to be more similar to myoepithelial cells than luminal epithelial cells. The manuscript would benefit from further discussion of the similarities and differences between the two cell types. For example, can the authors show that the 'cell crisis' occurring after prolonged culture of dCas9 3A3L targeted cells is similar to agonescence? The authors identify the repression of p14 as a required step toward the proliferation phenotype of the myoepithelial cells. However, it is worth noting that in vHMEC the TSS associated with p14 remains unmethylated and transcribed (Locke and Clark PMID: 23168266).

8. Clinical breast tumours are heterogenous and divided into a number of subtypes. Although some ER-ve/basal-like tumours display myoepithelial-like characteristics the current consensus is the vast majority of breast tumours arise from the luminal epithelial lineage rather than the myoepithelial lineage (eg see Molyneux et al 2010 PMID: 20804975). The use of myoepithelial cells therefore potentially detracts from the relevance of the findings to breast tumours in vivo (although as the authors suggest there is still debate over these points). Can they provide some evidence to show that the genes they study are indeed methylated in breast tumours in vivo? Currently they refer to previous studies which are either small-scale or based on methylation specific PCR which is very prone to false positive results. This is would be especially relevant for p14 given the authors focus on the need for dual methylation of p14/p16 in their system. Abundant quantitative data are available for this purpose (eg many 100s of samples are available from TCGA). Note: references 36 and 37 (Boulay et al 2012 and Wang et al 2014) are cited as providing evidence of HIC1 methylation in breast tumours but do not appear to contain any such data.

9. Similarly, previous work demonstrates that the majority genes that are frequently hypermethylated in breast carcinogenesis are repressed in the putative cell of origin (Sproul et al PNAS 2011, PMID: 21368160). Can the authors check the expression profile of the genes they target in the different mammary lineages in vivo and discuss their findings with relation to methylation of these promoters being a potential driver of carcinogenesis?

10. The authors conclude both p16 and 14 methylation are required to escape from senescence. However, that cells which escape senescence are indeed methylated at both loci, only that the bulk mean methylation is higher at both loci. Can they undertake an analysis of escapee clones or targeted single cell bsPCR to show that this is indeed the case?

11. At present only one result is supported by samples from multiple donors (Figure 4A). How robust are the alterations in expression and methylation observed in independent donors? Is the requirement for both p14 and p16 targeting seen in other donors?

12. The maintenance of DNA methylation 10 days post transfection in dCas9 3A3L cells that exhibit the proliferative phenotype is a relevant data (Supplementary Figure 16, results from the EpicArray) and it could have more visibility.

Minor points:

Time post transfection, passages and population doublings all used in the manuscript which makes it confusing to interpret the relationship of different experiments. The authors should be consistent in describing time elapsed during their experiments.

When mentioning the possible spontaneous DNA methylation caused by dCas9 3A3L, authors state "there was no identifiable difference in the phenotype between 3A3L and 3A3L Δ transfected cells as described later." (p 5). Not clear what they refer to.

In Figure 6, discordance in colour code between the legend and the figure.

It is unclear what the red boxes are in Supp Fig 10.

Add overall % methylation for bsPCR figures (eg Fig 3B).

Fig 3E – is the significance reported versus untransfected cells? Is average the mean or median?

Figure 4A – labels are incomplete.

Presentation of colorimetric growth assay data do not show variation observed in the 3A3L-delta samples (eg 4A and Fig 7). It is difficult therefore to conclude that the 3A3L samples do indeed proliferate to a greater extent. There is also no test of significance for these assays. Is the average proliferation mean or median?

Figure 6D – what are TP53-001 and TPE53-201?

p4 lines 132-133 typo "prior to before transfection"

p8 line 329 typo "p16 hypermethylated"

Reviewer #2 (Remarks to the Author):

The primary goal of this project is to demonstrate that methylation can be a driving event to

silence p16 and affect senescence bypass in epithelial cells. This project uses a relatively new approach to target methylation to the p16 locus using a dCas9-DNMT3A-3L fusion protein. The authors demonstrate that p16 can be silenced by this construct and that treated cells have increased proliferation. As outlined in my comments below, I am not entirely convinced based on their data that p16 methylation is the driving force behind their observations without additional experiments and controls. It is also unclear whether they can truly narrow this down given the large number of off-target methylation events they observe. Further I have a lingering concern about whether the events they observe are biologically meaningful, which I think limits my enthusiasm about the potential impact of this work. Even if p16 can be silenced by methylation, it doesn't preclude that in actual cells p16 methylation occurs after silencing.

Major comments

(1) I am concerned that the primary control used in this experiment is a misfolded DNMT3A-3L protein as opposed to a point mutant that would inactivate DNMT3A. As the authors note, prior work has sometimes observed a CRISPRi effect due to binding of the inactive group. The author's claim they do not observe this and that misfolding does not affect dCas9 folding or binding. First, the authors need to provide data to support (perhaps with active Cas9) that their construct does not affect Cas9 localization.

(2) Secondly, they need to repeat their experiments with a dCas9-DNMT3A-3L with point mutant control that inactivates the DNMT3A domain. I am concerned that the misfolded DNMT3A-3L construct no longer interacts with DNA the same way that DNMT3A-3L WT does and this will alter how much steric hindrance the protein confers both due to the DNMT3A-3L interaction as well as through the many additional factors that can be recruited by the functional DNMT3A-3L complex.

(3) Thirdly, I am concerned that one methylation-independent explanation is that the DNMT3A-3L construct can recruit chromatin remodelers that alter chromatin structure, independent of the observed DNA methylation and that these are the primary driver behind p16 silencing and not the observed DNA methylation. There is some evidence that chromatin remodeling precedes DNA methylation to silence p16 (Hinshelwood et al. Hum Mol Gen. 2009).

(4) The authors further need to make clear that while they show methylation can be a driver, it is unclear what happens physiologically. The discussion was lacking discussion of papers from Sue Clark for example that suggest that in HMECs methylation occurs after p16 silencing, not before.

(5) The language used to describe time in the paper and figures are confusing. At points the terms cell doublings, passage number, days, and hours post transfection are all used. The paragraph on line 221 page 6 is emblematic where nearly all of these terms are used over just a few sentences. It would be clearer if things could be compared on even terms (i.e. maybe stick to only days to refer to time and doublings to refer to growth).

(6) Surprisingly, the authors do not present data to show their cells undergo senescence, but only describe how their control cells undergo this process. The authors should present data directly showing that their control cells enter stasis and that the DNMT3A-3L cells bypass (e.g. plot days on x-axis and doublings on y-axis). This would particularly be helpful for their discussion of pictures or data taken at different time point when different cells are either still growing or have stopped.

(7) I do not find the data from PTEN methylation to be convincing. To my eyes, the gain in methylation at the PTEN promoter appears to be similar to that found at RASSF1A. Yet the gain in RASSF1A methylation is important, and the gain at PTEN is not. I.e. the distinction seems to be more about arbitrary statistical cutoffs, rather than a biologically meaningful distinction.

(8) The choice of a p-value cutoff at 0.0013 for DMR finding does not seem well justified to me. Typically, one chooses a p-value threshold prior to the study, rather than pick the threshold such

that all expected positive genes are positive. Another option would be to validate a subset of DMRs using bisulfite sequencing to guide a p-value choice.

(9) The authors should perform an overexpression control with a non-targeted sgRNA throughout. Without this, it is unclear as to whether the authors have observed specific methylation of the p16 locus by their construct, or whether this is caused by overexpression of 3A-3L in their cells. Redoing experiments with a functional DNMT3A-3L and sgRNAs targeted elsewhere would be more convincing.

(10) I agree with the authors that it is surprising that p14 and p16 silencing are both required in their cells. In the discussion, the authors mention several references that knockdown p16 only and induce senescence and say it is unclear whether they also knocked down p14ARF. As far as I can tell from aligning the provided siRNA sequences in Haga et al, they use p16-INK4A specific siRNAs that do not target the mRNA from p14ARF. This would indicate that p16 silencing is sufficient in this study. Regardless, if the authors want to conclude that p14 and p16 silencing are both required, I think they need to validate this finding using siRNA or another method. I think this is especially crucial since there are a large number of off-target effects (>9000 DMRs) from their methylation results.

(11) I was confused by what was meant of low and high cell density vs confluency. I think of these things as synonyms, but clearly the authors use these terms with different meaning. In part, I think the scales in Figs 4e and Supplementary Fig 10 make seeing the difference difficult.

(12) I did not find the PCA analysis in Figure 5c to be very convincing. This seems like a very indirect way to measure something that can be measured directly. I.e. take the down-regulated genes between DNMT3A-3L and ctrl targeted cells and ask using GSEA whether this gene set is in general have low expression in early passage cells and then do the opposite for up-regulated genes. Or look directly at the number of genes that return to the prior state. These would be much more useful numbers rather than reporting the % of variance. And these yield statistics to control for the probability of overlaps by chance.

(13) I am concerned about the large number of off-target effects (>9000 DMRs), and whether this prohibits one from concluding that p16/p14 targeted methylation is sufficient to bypass senescence. I'm wondering if the authors should perform a "rescue" type experiment to eliminate this concern. I.e. target dCas9-TET fusion to p16 locus to show reactivation, and that the cells then senesce. My concern would be that so many of the 9000 DMRs would also demethylate that again one could not limit the effect.

Minor Comments

(1) The figures need to be cleaned up in many places as they were difficult to interpret. For example: Scale bars are missing from the Me-DIP data in Figure 3; in Figure 3 the fonts are so small as to be illegible; the legend from Fig 4 a,b,c should be removed since only a single entry. Similar issues plague much of the supplement as well.

(2) The way the expression data is plotted in Fig 3e and 6 is confusing. I think it would be helpful to see the actual RPKM values for both early and late passage and comparisons with the different controls, rather than only fold changes. Or alternatively replot the RPKMs in a different format to show that they are significantly altered in early and late passages, and then use fold-change plots to indicate differences between dCas9 constructs.

(3) Also, in Fig. 6a I find the fact that the fold-change for RASSF1A-C is significant to be surprising given that the fold change looks to be less than 10%. Maybe I don't understand what's plotted here after all.

Reviewer #3 (Remarks to the Author):

In this study by Saunderson et al, the authors use CRISPR/dCas9 technology to epigenetically edit breast epithelial cells to increase DNA methylation. This clever system is employed to find alterations in promoters that permit bypass of "senescence" of these primary cultures. They find that genes that are commonly methylated in cancer, including RASSF1 and CDKN2A, can be repressed and thereby result in growth for a limited period of time. They suggest that these epigenetic manipulations therefore may be a potential first step during tumor cell escape from growth inhibition.

Overall, the data are interesting, however there are a number of questions/clarifications that need to be addressed. Firstly, as a non-specialist, I found the text to be terribly difficult to follow and the figures did not help with this confusion. There was a need to re-read many sections to understand the purpose the experiments, the result and the conclusion. Furthermore, the figure legends are not very clear in what was done. Also, the figures appear to be put together from various software programs and the resolution/sizing of text is hard to discern. With these thoughts in mind, this manuscript could greatly benefit from an extensive and thorough re-working of the text/figures. Secondly, although the methodology is interesting, it is not a novel approach nor is it clear what advance has been made here. The results are believable, but it seems more like a proof-of-principle manuscript than anything. The idea that p16/p14 impose an early proliferative arrest that can be overcome to promote growth is not terribly surprising. The second cell cycle arrest that the authors observe is extremely interesting, but is not followed up at all. What is the mechanism at play in this scenario? Thirdly, the statements about the "senescence escape" or "bypass of senescence" are not really supported by the data (see Major Point 1).

MAJOR POINTS

1. Notion of "senescence bypass" by epigenetic targeting.

- a. The experiments to make these conclusions would involve taking "established senescent cells" and then performing their various epigenetic interventions. From the experimental approach employed, they are interfering with cellular processes during the acquisition of senescence, not after senescence has been "established". This may seem like a semantic argument, but it is extremely critical. Others in the field have suggested that senescence can be bypassed by the accumulation of mutations in vitro, however it is also entirely possible that the outgrowth of cells that occurs with time results from a small fraction of cells that never were truly senescent that now have over-grown the remainder of the culture because senescent cells have no proliferation potential (which may also explain why these targeted cells cluster closer to the transcriptional profile of early passage cells). None of the experiments shown are supportive of a bypass of senescence, which is a critical flaw and conclusions need to be tempered around this issue.
- b. The in vitro experiments suggest that CRISPR/Cas9 3A3L targeted cells display higher rates of proliferation are not necessarily suggestive of increased susceptibility to tumors (which is what they argue in the introduction for the importance of targeting these regions). Additional tests for transformative potential, including the use of xenografts and tissue culture systems, would greatly expand on the understanding of the mechanisms at play in these cells.
- c. A potential explanation for the hypermethylation status of p16/p14 in human tumors is not discussed. How do the authors propose that this happens in this context?
- d. What is the second barrier that the "rejuvenated" clones encounter? Does p16/p14 expression get restored somehow? If so, would re-expression of their targeting system be sufficient to overcome this second obstacle? If so, are these cells even more proliferative?
- e. The generalizability of the cell lines is unclear. Are the results relevant for epithelial cell types only or other types as well?

2. Usefulness of the transient epigenetic modifications.

a. 5 days after transient targeting, the methylation status was altered (Fig. 3), yet there was only a modest change in gene expression. Since this is the case, it is hard to judge the importance of these transient changes. Furthermore, upon inspection of MeDIP profiles, it appears that a number of areas that are not targeted by the guides used for CRISPR targeting are altered, suggesting that some of the effects the authors see on their cell lines may be due to targeting of transcripts other than those designed to be influenced.

b. The data about seeding density influencing growth rates is very interesting, yet not well characterized. Earlier timepoints (to demonstrate what high vs low seeding looks like) should be used. Furthermore, a reason why the highly dense cells are dying needs to be explored. If cells are truly "senescent", they irreversibly growth arrested and able to stay in that state for prolonged period of time. The fact that they are dying indicates that the cells are actually highly stressed and NOT senescent.

MINOR POINTS

- Some of the figures are hard to discern, especially when taken from programs (is Supp. Fig 6 is essentially uninterpretable) and should be improved.

We are happy to know that all three reviewers find our study interesting and appreciate its importance in addressing the separation of cause and consequence in cancer epigenetics studies using epigenetic editing tools. We believe our study is essential to start understanding the functional contribution of epigenetic mechanisms to cellular physiology in general and tumourigenic phenotype in particular.

In the attached resubmission, we have addressed the criticisms raised by the referees and provide a large amount of new data to address these points. Following Referee 1 and Referee 3's advice, we have re-written the text and re-organised parts of the figures to make the manuscript easier to follow for non-specialists. A common major criticism was the role of the off-target effects in the proliferative phenotype we observe following transfection of primary cells with the CRISPR/dCas9 epigenetic editing tools. We now present new genome-wide DNA methylation data using Illumina Infinium MethylationEPIC BeadChip arrays to replace the previous MeDIP data; we have transfected primary cells with 3A3L without any gRNAs or various combinations of gRNAs to assess the driving role of *CDKN2A* methylation. Importantly, we have added new control experiments using the catalytic mutant of the 3A3L construct and dCas9-Dnmt3a alone too.

A second common criticism was that we didn't provide evidence that the 3A3L-delta targeted cells undergo senescence. We show now these cells stain positive for β -galactosidase and stop proliferating indicating that these cells undergo senescence.

In addition we have done similar experiments in primary luminal cells, which are harder to work with, and demonstrate that the phenotype we observe is similar in this primary cell type too.

Reviewers' comments:

Reviewer #1 (Remarks to the Author):

In this manuscript Saunderson et al report the successful use of the fusion protein dCas9 DNMT3A/3L to direct DNA methylation and repression of putative tumour suppressor genes in primary myoepithelial cells. They observe that DNA methylation targeted cells have increased proliferation compared to control cells, bypass senescence, and present a subset of genes with an expression profile similar to the one of early passage cells. Finally, the authors identify that p16 and p14 repression drive the senescence bypass.

It is well known that epigenetic dysregulation is a hallmark of cancer, but so far evidence that DNA methylation alterations can drive cancer has been correlative and indirect. For this reason, the work presented in the manuscript is of particular interest, as it shows, for the first time to the knowledge of the reviewer, that manipulation of DNA methylation at tumour suppressor gene promoters leads to an abnormal proliferative phenotype.

Although the data presented are fairly convincing, the manuscript would greatly benefit from further

work to robustly support the claims made. In addition parts of the manuscript are confusing and it contains a large number of typos. It therefore, requires clarification and editing.

We would like to thank the referee for the constructive criticism, we have made significant changes to the manuscript; added new data, reformulated confusing parts having colleagues outside our area of expertise read it and corrected typos.

Major points:

1. The authors describe initially the use of dCas9 3A3L to target DNA methylation in SKOV3 and 1089 cell lines (Fig.1). The successful use of this tool in cell lines has been reported previously (Stepper et al PMID: 27899645) and their results in these cell lines differ from those in primary myoepithelial cells (it is transient and limited to single genes). This difference is not followed up in the rest of the manuscript. Therefore, this initial part of the manuscript appears redundant and of marginal interest.

While we agree with the referee that the SKOV3 data is redundant, and have removed this from the manuscript, we believe that it is useful to mention how we established the transfection protocol in myoepithelial 1089 cell line, and we therefore kept this in the text but moved the figures to Supplementary Figure 1. We think it's useful for the reader to appreciate the similarities and differences in epigenetic editing between cell lines and primary cells.

2. The authors suggest the genes they target are not methylated in myoepithelial cells by examining of Roadmap Epigenomics data. However, this is a single replicate and they do not confirm this is the case for their myoepithelial cell cultures.

We agree with the referee that these figures were not sufficient or suitable. We have now included the genome-wide methylation data from donor 1 and 2 early passage myoepithelial cells (Fig. 4a) and the targeted bisulfite sequencing data from donor 1, 2 and 3 also from early passage cells for the panel of four genes (Fig. 4c-f), showing that these gene promoters are unmethylated in the primary myoepithelial cells we used in the study.

3. A major concern of epigenome-editing techniques is the contribution of off-target effects on the phenotype observed. From the MeDIP-seq results it is clear that dCas9 3A3L transfection into primary myoepithelial cells results in the methylation of many genomic regions other than the ones targeted by their cocktail of gRNAs. The authors suggest these off-target events are not caused by gRNA dependent dCas9 3A3L recruitment because they overlap minimally with computationally predicted off-target binding sites. Their evidence is dependent on the accuracy of the computational algorithm used (in this case Hsu et al 2013 from the Zhang lab). However, given that the prediction of CRISPR off-target binding events is in its infancy, the use of single, older algorithm only provides weak evidence towards this conclusion. The alternative, proposed by the authors is that dCas9 3A3L

is able to methylate genomic regions independently of its gRNAs. The best control to rule out an involvement

of these aspecific methylation events in senescence escape would be to transfect primary myoepithelial cells with dCas9 3A3L but no gRNAs, before analysing their proliferation. The experiments presented in Fig. 7a-b provide some evidence that this would not be the case. However, no methylation data are shown for these experiments. Comparison of methylation profiles resulting from dCas9 3A3L transfections with and without gRNAs would help clarify this issue. Similarly, ChIP for dCas9 3A3L with and without sgRNAs would provide further evidence to determine whether these off-target events are gRNA dependent or independent. Note, the nomenclature used by the authors here is slightly confusing as regions methylated by dCas9 3A3L independently of gRNAs are strictly still off-target events. In the manuscript the authors reserve off-target for predicted gRNA dependent sites. They should use gRNA dependent and independent off-target effects or similar terminology.

Off-target effects following dCas9-Dnmt transfection, considering the phenotype observed, are an important point raised by the referee. We have included a number of new experiments to address this. It is known from previous studies that Cas9 nucleases can generate off-target effects, but the assessment of off-targeting (gRNA dependent or independent) after Cas9-directed DNA methylation deposition is not well established in the literature and our work is the first study addressing it in significant detail. We discuss the general issue of off-targets more thoroughly with our new data outlined in subsequent points, but here the referee's question is whether off-targets are responsible for the phenotype observed. We show evidence that transfecting primary cells with dCas9 3A3L without gRNAs does not result in proliferation (Figure 3e, blue bars) therefore any unspecific gRNA independent effect caused by dCas9 3A3L is not responsible for the phenotype. Additionally, similar to siRNA experiments where specificity is questioned, we transfected primary cells with different combinations of two gRNAs targeting *CDKN2A* (assuming that these combinations will have distinct off-target effects because they are different gRNA sequences) and we observed that both combinations result in the proliferative effect (Figure 6g). Considering that targeting individual genes, *HIC1*, *PTEN* and *RASSF1A* with their own gRNAs and off-target effects, does not induce proliferation of the primary cells (Figure 6a,b) we are confident that targeting *CDKN2A* is the cause the proliferative phenotype. We modified the main text including all of the new data, and included a paragraph in the discussion clarifying the issue of off-targets.

4. From the data presented in Supp Fig 7, the magnitude of methylation change at the *PTEN* promoter looks similar to *HIC1*, *RASSF1* and *CDKN2A*. However, the authors suggest it does not gain methylation. We note that the authors empirically set the p-value threshold for differential methylation based on one locus, *HIC1*. How robust is the discrimination of differential methylation to false positives, can they validate with further loci using an independent assay?

We have thoroughly addressed this point now raised by the other two referees as well.

Due to the sequence biases associated with DNA immunoprecipitation using 5mC antibody, the non-linear quantification and relative measurements required by MeDIP we decided to analyse off-target

effects with a better assay (EPIC methylation arrays) which gives an absolute methylation value and more reliable data interpretation. We performed this analysis 10 days after transfection, rather than 5 days as for the MeDIP-seq, to allow more time for any targeted hypermethylation to be deposited and maintained in the cells, and also because we would not recover sufficient DNA needed for this assay at earlier timepoints due to the low number of primary cells we are working with. We include the data in Figure 2 and show that the target genes are significantly hypermethylated, with the greatest number of sequential hypermethylated probes. From the EPIC data, the *PTEN* and *KLLN* gene promoters are significantly hypermethylated (Figure 2b), although fewer probes cover the *PTEN* promoter.

Notably, we observe similar effects in 3 independent biological replicates, as in the MeDIP data, that there are a proportion of CpGs which become hypermethylated (946 probes excluding the on-targets, Figure 2e and f, Supplementary Data 1). The PCA analysis of the EPIC data also shows that the dCas9 3A3L and delta samples cluster separately (Supplementary Figure 11). It is possible that these are off-target gRNA-dependent or -independent effects that have been maintained in the cells, but we also cannot rule out that these occur as a result of the changing proliferative phenotype in these cells caused by the on-target DNA methylation since these cells have changed significantly from both 3A3LΔ and early passage cells (Figure 5, RNA-seq data). From the 946 >20% consistently hypermethylated probes, only two genes aside from the target gene panel contained more than 3 hypermethylated probes, indicative of DNA methylation spreading within the promoter region: *GOS2* and *C10orf41* (Supplementary Fig. 4a, b), potentially affecting transcription, although *C10orf41* is not expressed in any of the conditions. *GOS2* is downregulated in 3A3L targeted cells, but this may be a biological effect based on data from Novak et al, PMID: 19509227, which we mention in the manuscript. We also point out that a caveat of EPIC array data is that it does not cover every CpG and bring together the above summary by stating that we cannot ultimately rule out that hypermethylation events may occur due to off-target gRNA-dependent or -independent effects, but if they do occur it seems very unlikely that they drive the phenotype we observe in these primary cells, based on the additional functional experiments we have performed, summarised in the response to point 3.

5. The authors notice that dCas9 3A3L target cells have increased proliferation rate compared to control cells dCas9 3A3LΔ, but that this phenotype is dependent on cell seeding density. However, the effect of the seeding density observed does not fit with the model proposed in the manuscript. According to the model, targeting of both p16 and p14 enables senescence bypass senescence. Targeted methylation is however only partially efficient and it is reasonable to suppose that hypermethylation of both p14 and p16 will occur in a small fraction of cells. The number of double-targeted cells would be expected to increase with the number of plated cells, ie the number of escapees and therefore overall proliferation rate should increase with seeding density. However, the authors describe the opposite effect, proliferation and, presumably, the number of escapees are higher with low seeding density.

As the seeding density is an important parameter in their experiments, it should be specified for the further experiments performed (eg are MeDIP and RNAseq data derived from cells plated at low or

high densities?). It would also aid future research in this area if cell-seeding densities used were specified (eg what are low, medium and high in Fig 4)?

This is an interesting point raised by the referee. We have now added experimental details indicating the number of seeded cells in low vs high density culture conditions (see Materials and Methods 'Proliferation assay' and in the main text p5/lines 191 – 194). We think there are two confounding phenomena that are contributing to the effect. Firstly, as the referee notes, transfection efficiency and cell numbers contribute to the number of successfully targeted cells. Secondly, we think that paracrine signalling at high density might inhibit myoepithelial cell proliferation in vitro. While this latter hypothesis is interesting and we will follow it up, it is beyond the scope of this manuscript. Also, this inhibition might be specific to myoepithelial cells as we did not observe this effect with luminal cells.

Based on these observations and the additional new data we have moved this data to the supplementary figures as it is not a key message of the paper.

6. The authors conclude that targeting of p14 + p16 by dCas9 3A3L enables escape from senescence in primary myoepithelial cells. However, they do not show that untransfected or dCas9 3A3L-delta cells undergo senescence. The reference given in the introduction for untransfected cells (Brenner et al 1998) discusses HMECs rather than primary myoepithelial cells isolated with the protocol used here. Can the authors provide data using markers to properly demonstrate that these cells do indeed undergo senescence?

We provide new data indicating that the cells transfected with the 3A3L Δ construct and untransfected cells undergoing senescence, evidenced by p16 overexpression (Figure 2g), inhibited proliferation, and β -galactosidase staining (Figure 3d).

7. As discussed by the authors, cultured HMEC cells enter senescence but a subpopulation can escape (vHMECs), before entering a final period of growth arrest which has been termed agonescence. The authors observe similarities between the growth phenotype of the dCas9 3A3L targeted myoepithelial cells and vHMECs. This would be expected as HMECs are believed to be more similar to myoepithelial cells than luminal epithelial cells. The manuscript would benefit from further discussion of the similarities and differences between the two cell types. For example, can the authors show that the 'cell crisis' occurring after prolonged culture of dCas9 3A3L targeted cells is similar to agonescence? The authors identify the repression of p14 as a required step toward the proliferation phenotype of the myoepithelial cells. However, it is worth noting that in vHMEC the TSS associated with p14 remains unmethylated and transcribed (Locke and Clark PMID: 23168266).

Our new data shows that overexpression of hTERT via stable integration in the methylation targeted cells provides a growth advantage to the cells, demonstrated by the increasing proportion of GFP^{+ve}

cells (Figure 3f) and overall increasing numbers of cells (Supplementary Figure 10d). This indicates that, at least in part, shortening telomeres are contributing to the late cell cycle arrest, showing a similarity between vHMECs and our cells. Our new data also show that methylating p16 is sufficient to enable proliferation (discussed in more detail below for point 10), we have included qPCR data showing p14 is indeed still expressed in these cells (Supplementary Figure 19b), similar to vHMECs. We referenced the paper in the manuscript, thank you for raising our attention to it.

8. Clinical breast tumours are heterogenous and divided into a number of subtypes. Although some ER-ve/basal-like tumours display myoepithelial-like characteristics the current consensus is the vast majority of breast tumours arise from the luminal epithelial lineage rather than the myoepithelial lineage (eg see Molyneux et al 2010 PMID: 20804975). The use of myoepithelial cells therefore potentially detracts from the relevance of the findings to breast tumours in vivo (although as the authors suggest there is still debate over these points). Can they provide some evidence to show that the genes they study are indeed methylated in breast tumours in vivo? Currently they refer to previous studies which are either small-scale or based on methylation specific PCR which is very prone to false positive results. This would be especially relevant for p14 given the authors focus on the need for dual methylation of p14/p16 in their system. Abundant quantitative data are available for this purpose (eg many 100s of samples are available from TCGA). Note: references 36 and 37 (Boulay et al 2012 and Wang et al 2014) are cited as providing evidence of HIC1 methylation in breast tumours but do not appear to contain any such data.

We have now used the Wanderer program which utilises 450K array DNA methylation data from the TCGA database to analyse the methylation status of the target genes in 798 invasive breast carcinomas compared to 98 normal breast samples (Supplementary Fig. 2) which shows these genes become hypermethylated in a subset of breast cancer cases.

We apologise about the wrong reference on HIC1 methylation. Separately, we have also included data targeting DNA methylation to CDKN2A in primary luminal cells (Supplementary Fig. 20) which shows similar senescence prevention. Please also see response to Referee 3 Point 1b for further discussion of luminal cells.

9. Similarly, previous work demonstrates that the majority genes that are frequently hypermethylated in breast carcinogenesis are repressed in the putative cell of origin (Sproul et al PNAS 2011, PMID: 21368160). Can the authors check the expression profile of the genes they target in the different mammary lineages in vivo and discuss their findings with relation to methylation of these promoters being a potential driver of carcinogenesis?

We consider that in our experimental system, primary myoepithelial cells are appropriate to analyse expression of these genes, relevant to the myoepithelial lineage. (Supplementary Fig 3a, discussed

on page 4/lines 140 - 142). We see low expression of *HIC1*, *RASSF1A* and p14, and higher expression of *PTEN* and p16, which is expected to increase in primary cell culturing *in vitro*. Our argument as to why even silent genes are relevant in initiating cancer is that gene expression may still be readily upregulated if the individual cells requires it, i.e. during cellular stress (as is the case for p16). As such, if a gene is repressed via more permanent DNA methylation then stress-induced upregulation of expression could be blocked and this could drive the initiating processes in carcinogenesis. We also include some discussion about possible prior silencing of p14/p16 on p11/lines 461 - 471.

10. The authors conclude both p16 and 14 methylation are required to escape from senescence. However, that cells which escape senescence are indeed methylated at both loci, only that the bulk mean methylation is higher at both loci. Can they undertake an analysis of escapee clones or targeted single cell bsPCR to show that this is indeed the case?

In the manuscript we have further analysed the p14/p16 individual or combined role in the phenotype observed (Figure 6). Due to the large number of conditions in the new experiments and availability of primary cells from the same donor, the magnetic sorting was avoided to reduce cell loss. Surprisingly, we observed that under these experimental conditions targeting p16 alone was sufficient to enable the proliferative phenotype in myoepithelial cells from 3 different donors (Figure 6d), and more variably even with p14 alone, although it appears that targeting both may enhance the outgrowth of myoepithelial cells (Figure 6f, Supplementary Fig. 18c).

We have performed targeted bisulfite sequencing of the targeted genes and shown that the percentage of some CpGs at p16 and p14 promoters are close to 100% methylated (Fig. 4f) indicating that while possibly dynamic, most cells do have both promoters methylated at the point of analysis.

11. At present only one result is supported by samples from multiple donors (Figure 4A). How robust are the alterations in expression and methylation observed in independent donors? Is the requirement for both p14 and p16 targeting seen in other donors?

We have now repeated the independent and double targeting of 14 and p16 in 3 donors (Fig. 6c-e) and observe that p16 targeting is sufficient to drive the proliferative phenotype in all donors. However, targeting DNA methylation to both p14 and p16 significantly increases the population doublings of these cells in donors 1 and 3 (Fig. 6f, Supplementary Fig. 18c) and targeting p14 can prevent senescence in donor 1 (Fig. 6c).

12. The maintenance of DNA methylation 10 days post transfection in dCas9 3A3L cells that exhibit the proliferative phenotype is a relevant data (Supplementary Figure 16, results from the EpicArray) and it could have more visibility.

This data is 38 days after transfection but is indeed an important data and we therefore moved this data to the main figures (Figure 4a, b – including a separate donor) and provide additional validation using targeted sequencing (Figure 4c – f).

Minor points:

Time post transfection, passages and population doublings all used in the manuscript which makes it confusing to interpret the relationship of different experiments. The authors should be consistent in describing time elapsed during their experiments.

We have simplified the wording used so that we only describe days after transfection. We use cumulative population doubling to show that 3A3LΔ targeted and untransfected cells do indeed enter senescence (Figure 3c) and for exploring the difference in growth profiles between p16- and p16/p14-targeted cells (Figure 6f).

When mentioning the possible spontaneous DNA methylation caused by dCas9 3A3L, authors state “there was no identifiable difference in the phenotype between 3A3L and 3A3LΔ transfected cells as described later.” (p 5). Not clear what they refer to.

We agree this sentence is ambiguous and have removed it.

In Figure 6, discordance in colour code between the legend and the figure.

We thank the referee for pointing this out and have corrected this issue.

It is unclear what the red boxes are in Supp Fig 10.

We have now described them in the Figure legend.

Add overall % methylation for bsPCR figures (eg Fig 3B).

We have added this for the HIC1 region. Our EPICarray data provides a better idea of the amount of DNA methylation deposited after transfection.

Fig 3E – is the significance reported versus untransfected cells? Is average the mean or median?

As the RNA-seq and qPCR data from 10 days post-transfection is a better reflection of our findings we have removed the qPCR data from day 5. Additionally, we have included mean \pm SEM where applicable in all figure legends. We have fully explained significance values in the figure legends.

Figure 4A – labels are incomplete.

These have been added.

Presentation of colorimetric growth assay data do not show variation observed in the 3A3L-delta samples (eg 4A and Fig 7). It is difficult therefore to conclude that the 3A3L samples do indeed proliferate to a greater extent. There is also no test of significance for these assays. Is the average proliferation mean or median?

We show that 3A3LΔ targeted cells have minimal growth after 15 days post-transfection (Figure 3c), therefore indicating that there is very little proliferation of these cells. Furthermore, Figure 3c shows there is minimal proliferation of 3A3LΔ targeted cells after 20 and 25 days post-transfection. We hope the reviewer agrees this highlights the greater 3A3L proliferation adequately. We have updated the figure legends and materials and methods to state the mean ± the SEM are plotted.

Figure 6D – what are TP53-001 and TPE53-201?

We have updated this figure to clarify that these are two alternative transcripts of p53.

p4 lines 132-133 typo “prior to before transfection”

We have corrected this issue.

p8 line 329 typo “p16 hypermethylated”

We have corrected the text.

Reviewer #2 (Remarks to the Author):

The primary goal of this project is to demonstrate that methylation can be a driving event to silence p16 and affect senescence bypass in epithelial cells. This project uses a relatively new approach to target methylation to the p16 locus using a dCas9-DNMT3A-3L fusion protein. The authors demonstrate that p16 can be silenced by this construct and that treated cells have increased proliferation. As outlined in my comments below, I am not entirely convinced based on their data that p16 methylation is the driving force behind their observations without additional experiments and controls. It is also unclear whether they can truly narrow this down given the large number of

off-target methylation events they observe. Further I have a lingering concern about whether the events they observe are biologically meaningful, which I think limits my enthusiasm about the potential impact of this work. Even if p16 can be silenced by methylation, it doesn't preclude that in actual cells p16 methylation occurs after silencing.

We want to thank the referee for the comments and questions, which we responded to below. Our work has value beyond p16, in separating cause and consequence in DNA methylation biology in general, demonstrating the importance of epigenetic modifications in instructing physiological changes and sets a precedence for further research in the epigenetics editing field therefore we believe this work will have significant impact to understanding epigenetics mechanisms in general and cancer epigenetics in particular.

Major comments

(1) I am concerned that the primary control used in this experiment is a misfolded DNMT3A-3L protein as opposed to a point mutant that would inactivate DNMT3A. As the authors note, prior work has sometimes observed a CRISPRi effect due to binding of the inactive group. The author's claim they do not observe this and that misfolding does not affect dCas9 folding or binding. First, the authors need to provide data to support (perhaps with active Cas9) that their construct does not affect Cas9 localization.

This is an important point raised by the referee, we have now provided new data using a point mutated version of dCas9 3A3L where the catalytic domain of the 3A is abolished by C706A amino acid substitution (Figure 3e), we repeated this experiment several times in Donor 1 and Donor 3 (Supplementary Fig. 7) with no proliferative effect. In addition, we included new controls such as transfecting cells with dCas9 3A3L without gRNA and targeting the catalytically inactive Cas9 with no effector domain with all gRNAs to exclude a CRISPRi effect (Figure 3e). All of the above did not result in the proliferative phenotype. Finally, we tested as well the dCas9-3A fusion with all gRNAs and our results show that this is sufficient to drive the proliferative phenotype in the absence of 3L (Figure 3e), even though work using cell lines has shown dCas9-3A deposits less DNA methylation than dCas9-3A3L (Stepper et al PMID: 27899645).

(2) Secondly, they need to repeat their experiments with a dCas9-DNMT3A-3L with point mutant control that inactivates the DNMT3A domain. I am concerned that the misfolded DNMT3A-3L construct no longer interacts with DNA the same way that DNMT3A-3L WT does and this will alter how much steric hindrance the protein confers both due to the DNMT3A-3L interaction as well as through the many additional factors that can be recruited by the functional DNMT3A-3L complex.

We addressed the 3A point mutant question as mentioned above.

(3) Thirdly, I am concerned that one methylation-independent explanation is that the DNMT3A-3L construct can recruit chromatin remodelers that alter chromatin structure, independent of the observed DNA methylation and that these are the primary driver behind p16 silencing and not the observed DNA methylation. There is some evidence that chromatin remodeling precedes DNA methylation to silence p16 (Hinshelwood et al. Hum Mol Gen. 2009).

As mentioned above, the data indicates that the proliferation phenotype can be observed even when using dCas9-3A construct alone and not when the point mutant is being used. We are therefore confident that in this setup it is the DNA methylation that is driving downstream processes and the proliferation phenotype. However, as mentioned in the discussion (p11/lines 461 - 471) we cannot exclude that cells which are permissive to dCas9 3A3L-targeted hypermethylation in our study, already have low or repressed p16 expression at that stage. We are following up the importance of heterogeneity within primary cell population, but our study demonstrates that DNA methylation of an unmethylated p16 (within a cell population, Fig. 4a,f) has a permanent effect in preventing p16 overexpression.

(4) The authors further need to make clear that while they show methylation can be a driver, it is unclear what happens physiologically. The discussion was lacking discussion of papers from Sue Clark for example that suggest that in HMECs methylation occurs after p16 silencing, not before.

We agree with the referee that *in vivo*, depending on the tissue and gene, upstream epigenetic mechanisms may drive silencing and DNA methylation consolidating the effect. As mentioned in response to point 3, we do not know if low/repressed p16 is a requirement for successful DNA methylation targeting. Nevertheless, our work is essential in the effort to start addressing and interpreting the functional role of epigenetic events in cancer and ageing context when too often opinions are divided on DNA methylation simply being an inert biomarker or functional player in the evolution of the disease. This applies to other epigenetic modifications too.

As the reviewer rightly points out, gene repression via histones may preclude DNA methylation *in vivo* and this study does not and cannot distinguish between whether either is more important for driving carcinogenesis in tissue. We have clarified this in the discussion (Discussion, p11/lines 461 - 463)

(5) The language used to describe time in the paper and figures are confusing. At points the terms cell doublings, passage number, days, and hours post transfection are all used. The paragraph on line 221 page 6 is emblematic where nearly all of these terms are used over just a few sentences. It would be clearer if things could be compared on even terms (i.e. maybe stick to only days to refer to time and doublings to refer to growth).

We have addressed this and kept to referring to days post-transfection.

(6) Surprisingly, the authors do not present data to show their cells undergo senescence, but only describe how their control cells undergo this process. The authors should present data directly showing that their control cells enter stasis and that the DNMT3A-3L cells bypass (e.g. plot days on x-axis and doublings on y-axis). This would particularly be helpful for their discussion of pictures or data taken at different time point when different cells are either still growing or have stopped.

We thank the referee for this constructive criticism, we included new data showing that cells undergo senescence (β -galactosidase staining, Figure 3d) while those that bypass it stain negative. We also provide population doubling data as suggested by the referee (Figure 3c).

(7) I do not find the data from PTEN methylation to be convincing. To my eyes, the gain in methylation at the PTEN promoter appears to be similar to that found at RASSF1A. Yet the gain in RASSF1A methylation is important, and the gain at PTEN is not. I.e. the distinction seems to be more about arbitrary statistical cutoffs, rather than a biologically meaningful distinction.

Please see our answers above as a similar concern was raised by Referee 1, point 4.

(8) The choice of a p-value cutoff at 0.0013 for DMR finding does not seem well justified to me. Typically, one chooses a p-value threshold prior to the study, rather than pick the threshold such that all expected positive genes are positive. Another option would be to validate a subset of DMRs using bisulfite sequencing to guide a p-value choice.

We give a detailed answer above, Referee 1 point 4. We have used a better method (the EPIC array) which gives much greater depth per CpG measured, to assess DNA methylation targeting and off-target effects. This was performed at day 10 post-transfection rather than rather than 5 days (as for the MeDIP-seq) to allow more time for any targeted hypermethylation to be deposited and maintained in the cells, and also to enable us to harvest enough cells and therefore genomic DNA due to the low number of primary cells we are working with and the relatively high DNA requirement for EPIC arrays.

(9) The authors should perform an overexpression control with a non-targeted sgRNA throughout. Without this, it is unclear as to whether the authors have observed specific methylation of the p16 locus by their construct, or whether this is caused by overexpression of 3A-3L in their cells. Redoing experiments with a functional DNMT3A-3L and sgRNAs targeted elsewhere would be more convincing.

When we target dCas9 3A3L using gRNA designed to a different genomic location than p16 (experiments where we use gRNA targeting *HIC1* or *PTEN* or *RASSF1* or *CDKN2A* CGI3 on their own)

we don't see the proliferative phenotype (Figure 6a,b). Considering our arguments above (Referee 1 point 3) we think the data is convincing that it is DNA methylation of p16 that drives the phenotype.

(10) I agree with the authors that it is surprising that p14 and p16 silencing are both required in their cells. In the discussion, the authors mention several references that knockdown p16 only and induce senescence and say it is unclear whether they also knocked down p14ARF. As far as I can tell from aligning the provided siRNA sequences in Haga et al, they use p16-INK4A specific siRNAs that do not target the mRNA from p14ARF. This would indicate that p16 silencing is sufficient in this study. Regardless, if the author's want to conclude that p14 and p16 silencing are both required, I think they need to validate this finding using siRNA or another method. I think this is especially crucial since there are a large number of off-target effects (>9000 DMRs) from their methylation results.

We have now thoroughly addressed this point above as answered to Referee 1 Point 3, 4 and 10. Briefly, we observe that p16 targeting is sufficient to drive the proliferative phenotype in all donors. However, targeting DNA methylation to both p14 and p16 significantly increases the population doublings of these cells in donors 1 and 3 (Fig. 6f) and targeting p14 can prevent senescence in donor 1 (Fig. 6c).

(11) I was confused by what was meant of low and high cell density vs confluency. I think of these things as synonyms, but clearly the authors use these terms with different meaning. In part, I think the scales in Figs 4e and Supplementary Fig 10 make seeing the difference difficult.

We agree that this text was confusing. We were trying to keep separate the terms describing the amount of cells seeded initially (low, medium high density seeding) vs the measurement of cell growth during the experiment. Due to our addition of other more relevant data in the resubmission we have reduced the focus on this part of the manuscript and moved the data to Supplementary Figure 6, we have also included values for the seeding density in the materials and methods 'Proliferation assay' and the main text p5/lines 191 – 194.

(12) I did not find the PCA analysis in Figure 5c to be very convincing. This seems like a very indirect way to measure something that can be measured directly. I.e. take the down-regulated genes between DNMT3A-3L and ctrl targeted cells and ask using GSEA whether this gene set is in general have low expression in early passage cells and then do the opposite for up-regulated genes. Or look directly at the number of genes that return to the prior state. These would be much more useful numbers rather than reporting the % of variance. And these yield statistics to control for the probability of overlaps by chance.

We have addressed this point now, shown in Supplementary Fig 15 and 16b.

(13) I am concerned about the large number of off-target effects (>9000 DMRs), and whether this

prohibits one from concluding that p16/p14 targeted methylation is sufficient to bypass senescence. I'm wondering if the authors should perform a "rescue" type experiment to eliminate this concern. I.e. target dCas9-TET fusion to p16 locus to show reactivation, and that the cells then senesce. My concern would be that so many of the 9000 DMRS would also demethylate that again one could not limit the effect.

Referee 1 has raised this concern as well and we addressed this as described above in point 3 and point 4.

Minor Comments

(1) The figures need to be cleaned up in many places as they were difficult to interpret. For example: Scale bars are missing from the Me-DIP data in Figure 3; in Figure 3 the fonts are so small as to be illegible; the legend from Fig 4 a,b,c should be removed since only a single entry. Similar issues plague much of the supplement as well.

Thank you for the constructive comments. We have addressed these issues now.

(2) The way the expression data is plotted in Fig 3e and 6 is confusing. I think it would be helpful to see the actual RPKM values for both early and late passage and comparisons with the different controls, rather than only fold changes. Or alternatively replot the RPKMs in a different format to show that they are significantly altered in early and late passages, and then use fold-change plots to indicate differences between dCas9 constructs.

We appreciate the referee's comments regarding the expression data. We have separated the p16 data from the rest to make the differences clearer in Figure 2g and h. We have also included additional gene expression data in the manuscript showing the expression level of our target genes at early passage in 3 donors (Supplementary Fig. 3a) and a comparison between early passage and dCas9 3A3L transfected cells 35-37 days post-transfection in the three donors to clarify the issue of changes in gene expression compared to early passage cells (Supplementary Fig. 13).

(3) Also, in Fig. 6a I find the fact that the fold-change for RASSF1A-C is significant to be surprising given that the fold change looks to be less than 10%. Maybe I don't understand what's plotted here after all.

We agree the graph does not necessarily look significant in the originally submission. Firstly, in our re-submission we have updated the figure as the original was not representative of the individual transcripts (Figure 2h). Secondly, the significance shown on the figure is based on the DESeq2 differential expression analysis package from R, which has its own normalisation and significance calculation method.

Reviewer #3 (Remarks to the Author):

In this study by Saunderson et al, the authors use CRISPR/dCas9 technology to epigenetically edit breast epithelial cells to increase DNA methylation. This clever system is employed to find alterations in promoters that permit bypass of “senescence” of these primary cultures. They find that genes that are commonly methylated in cancer, including RASSF1 and CDKN2A, can be repressed and thereby result in growth for a limited period of time. They suggest that these epigenetic manipulations therefore may be a potential first step during tumor cell escape from growth inhibition.

Overall, the data are interesting, however there are a number of questions/clarifications that need to be addressed. Firstly, as a non-specialist, I found the text to be terribly difficult to follow and the figures did not help with this confusion. There was a need to re-read many sections to understand the purpose the experiments, the result and the conclusion. Furthermore, the figure legends are not very clear in what was done. Also, the figures appear to be put together from various software programs and the resolution/sizing of text is hard to discern. With these thoughts in mind, this manuscript could greatly benefit from an extensive and thorough re-working of the text/figures. Secondly, although the methodology is interesting, it is not a novel approach nor is it clear what advance has been made here. The results are believable, but it seems more like a proof-of-principle manuscript than anything. The idea that p16/p14 impose an early proliferative arrest that can be overcome to promote growth is not terribly surprising. The second cell cycle arrest that the authors observe is extremely interesting, but is not followed up at all. What is the mechanism at play in this scenario? Thirdly, the statements about the “senescence escape” or “bypass of senescence” are not really supported by the data (see Major Point 1).

MAJOR POINTS

1. Notion of “senescence bypass” by epigenetic targeting.

a. The experiments to make these conclusions would involve taking “established senescent cells” and then performing their various epigenetic interventions. From the experimental approach employed, they are interfering with cellular processes during the acquisition of senescence, not after senescence has been “established”. This may seem like a semantic argument, but it is extremely critical. Others in the field have suggested that senescence can be bypassed by the accumulation of mutations in vitro, however it is also entirely possible that the outgrowth of cells that occurs with time results from a small fraction of cells that never were truly senescent that now have over-grown the remainder of the culture because senescent cells have no proliferation potential (which may also explain why these targeted cells cluster closer to the transcriptional profile of early passage cells). None of the experiments shown are supportive of a bypass of senescence, which is a critical flaw and conclusions need to be tempered around this issue.

We appreciate the constructive comment, indeed the wording “escape” was incorrect as it is very likely these cells never entered senescence. We have included new data, as requested by all referees, showing that 3A3LA and untransfected cells indeed enter senescence (Figure 3d). The β -galactosidase staining shows that in the proliferating groups of cells are staining negative for β -

galactosidase, indicating that senescence was prevented in this case. We have also adjusted the wording throughout the paper to 'prevent' rather than 'escape' senescence.

b. The in vitro experiments suggest that CRISPR/Cas9 3A3L targeted cells display higher rates of proliferation are not necessarily suggestive of increased susceptibility to tumors (which is what they argue in the introduction for the importance of targeting these regions). Additional tests for transformative potential, including the use of xenografts and tissue culture systems, would greatly expand on the understanding of the mechanisms at play in these cells.

That is correct and we have undertaken experiments, such as anchorage independent growth assay, which can inform on transformative potential of the 3A3L edited cells. When we did this assay with myoepithelial cells we did not see outgrowth of cells (Supplementary Figure 9) therefore we could not ethically justify the more stringent assay such as using xenograft. As a result of this we do not argue that the myoepithelial cells are transformed. However, we have now extended our study by using luminal cells, which can form colonies in 3D and grow after DNA methylation targeting to *CDKN2A*. We believe that a more thorough assessment of phenotypic change of luminal cells deserves an extensive analysis so we will follow this up in a different study.

c. A potential explanation for the hypermethylation status of p16/p14 in human tumors is not discussed. How do the authors propose that this happens in this context?

Please see response to Referee 1 point 8 and 9. We have included in the discussion points about how p14/p16 may be repressed via histone modifications prior to DNA hypermethylation (p11/461 – 471).

d. What is the second barrier that the "rejuvenated" clones encounter? Does p16/p14 expression get restored somehow? If so, would re-expression of their targeting system be sufficient to overcome this second obstacle? If so, are these cells even more proliferative?

Please see referee 1 point 7, we have included new data regarding this question. Additionally, we include measurements of p16 expression during the hTERT experiment and a separate experiment (Supplementary Fig. 10b and c), showing that p16 does not become re-expressed.

e. The generalizability of the cell lines is unclear. Are the results relevant for epithelial cell types only or other types as well?

See comments above, point 1b.

2. Usefulness of the transient epigenetic modifications.

a. 5 days after transient targeting, the methylation status was altered (Fig. 3), yet there was only a modest change in gene expression. Since this is the case, it is hard to judge the importance of these transient changes. Furthermore, upon inspection of MeDIP profiles, it appears that a number of areas that are not targeted by the guides used for CRISPR targeting are altered, suggesting that some of the effects the authors see on their cell lines may be due to targeting of transcripts other than those designed to be influenced.

This is an important point from all reviewers and has been address in Referee 1 point 3, 4 and 9

b. The data about seeding density influencing growth rates is very interesting, yet not well characterized. Earlier timepoints (to demonstrate what high vs low seeding looks like) should be used. Furthermore, a reason why the highly dense cells are dying needs to be explored. If cells are truly “senescent”, they irreversibly growth arrested and able to stay in that state for prolonged period of time. The fact that they are dying indicates that the cells are actually highly stressed and NOT senescent.

We thank the reviewer for this comment, and ask them to refer to Reviewer 1 point 5 for a thorough discussion and explanation.

MINOR POINTS

- Some of the figures are hard to discern, especially when taken from programs (is Supp. Fig 6 is essentially uninterpretable) and should be improved.

We have significantly reworked the figures, in particular we have removed Supplementary Figure 6 and replaced it with targeted bisulfite sequencing showing hypomethylation of our target genes in the primary myoepithelial cells from the three donors used in this study (Figure 4c - f).

REVIEWERS' COMMENTS:

Reviewer #1 (Remarks to the Author):

This revised manuscript by Saunderson et al provides substantial additional data that have addressed the concerns regarding their original submission. The authors now convincingly demonstrated that transient targeting of DNA methylation to the CDKN2A locus in primary myoepithelial cells enables them to bypass senescence, a highly important finding. In particular the authors present additional data demonstrating that their results are not explained by off target effects and that primary myoepithelial cells do indeed undergo senescence in their experiments.

The only point that was not adequately addressed was the analysis of DNA methylation data in primary breast tumours (point 8 in the previous round). The authors, present an analysis of data from the Cancer Genome Atlas to suggest that their targeted genes are hypermethylated in primary breast tumours. However, the way these data are visualised means that this point is not apparent from Supplementary Figure 2, particularly in the case of CDKN2A. Perhaps the data can be plotted in a clearer and simpler way, for example heatmaps similar to those in Figure 4.

Reviewer #2 (Remarks to the Author):

I appreciate the time and effort expended on the additional experiments and changes to the manuscript. These have addressed my original concerns. I had a few minor comments concerning the new data:

- 1) The authors should label the p16 and p14 promoter on Fig S2 and need to include statistics.
- 2) Page 9 line 356: "... resulted in significantly more proliferation in ..." No p-value is provided, but should be.
- 3) The authors should comment on the variability in Supplementary Fig. 11. The text on p7 simply says they tracked it over time. But the graph indicates huge differences between the donor cells at late passage. And that the 3A3Ldelta control cells look nothing like the late-passage cells in terms of methylation.
- 4) p.4 line 159. If the authors want to make a point about spreading they need to provide the full analysis. e.g. How many genes have probe densities high enough to detect spreading? Without knowing this, it is hard to comment one way or another on whether spreading is frequent or an isolated event. It would likely be more appropriate though to provide an analysis of the autocorrelation between hypermethylation events and distance between CpG sites.
- 5) p.4 line 164. I found the statement here overly strong, redundant, and a bit out of place, since it refers to data later in the paper. I would just remove. I think the statement in the discussion in page 10, line 423 is sufficient.

Reviewer #3 (Remarks to the Author):

The manuscript is improved in this revised version. My comments have been satisfactorily addressed. However, some areas are still a little difficult to understand and can be further improved with copyediting. Below are a few suggestions from just the first page:

In some instances, it would be better to use "inability to engage senescence arrest" rather than "prevented from entering senescence" (abstract, lines 42-43).

The last sentence of the abstract is confusing. A modification more appropriate would be, "This work demonstrates that hit-and-run epigenetic events can prevent senescence arrest, which may facilitate the early stages of tumor initiation" or something similar.

The third "Highlight" bullet point suggests hyper-proliferation is driven by p16 when that is clearly the opposite of the actual truth.

REVIEWERS' COMMENTS:

Reviewer #1 (Remarks to the Author):

This revised manuscript by Saunderson et al provides substantial additional data that have addressed the concerns regarding their original submission. The authors now convincingly demonstrated that transient targeting of DNA methylation to the CDKN2A locus in primary myoepithelial cells enables them to bypass senescence, a highly important finding. In particular the authors present additional data demonstrating that their results are not explained by off target effects and that primary myoepithelial cells do indeed undergo senescence in their experiments.

The only point that was not adequately addressed was the analysis of DNA methylation data in primary breast tumours (point 8 in the previous round). The authors, present an analysis of data from the Cancer Genome Atlas to suggest that their targeted genes are hypermethylated in primary breast tumours. However, the way these data are visualised means that this point is not apparent from Supplementary Figure 2, particularly in the case of CDKN2A. Perhaps the data can be plotted in a clearer and simpler way, for example heatmaps similar to those in Figure 4.

We are glad the reviewer found the revised manuscript to be convincing and thank them for their additional comment. Since the 450K array has very poor coverage of CDKN2A, we found that plotting the data as a heat map did not help to clarify the data and support our point. To address this in a different way we have added labelling for p14 and p16, included the mean beta value of the probes for CDKN2A and shown which probes are significantly hypermethylated in breast carcinoma compared with control for all the genes. We hope this, in conjunction with the references we have given showing evidence of CDKN2A hypermethylation in cancer, is sufficient evidence for our argument that hypermethylation of these genes is found in some breast tumours.

Reviewer #2 (Remarks to the Author):

I appreciate the time and effort expended on the additional experiments and changes to the manuscript. These have addressed my original concerns. I had a few minor comments concerning the new data:

1) The authors should label the p16 and p14 promoter on Fig S2 and need to include statistics.

We are happy the reviewer feels we have address the original concerns with our new data. See the answer to Reviewer 1, we hope this has made the Figure sufficiently clear.

2) Page 9 line 356: "... resulted in significantly more proliferation in ..." No p-value is provided, but should be.

We have now added the statistical analysis in the main text relating to the two-way ANOVA in Supplementary Fig. 18c.

3) The authors should comment on the variability in Supplementary Fig. 11. The text on p7 simply

says they tracked it over time. But the graph indicates huge differences between the donor cells at late passage. And that the 3A3Ldelta control cells look nothing like the late-passage cells in terms of methylation.

We agree with the reviewer and have added the following to the main text: 'Clustering all EPIC array data using principle component analysis (PCA) showed that the dCas9 3A3L and 3A3LΔ targeted cells clustered separately and that late passage cells have greater variability in the global DNA methylation pattern (Supplementary Fig. 11).'

4) p.4 line 159. If the authors want to make a point about spreading they need to provide the full analysis. e.g. How many genes have probe densities high enough to detect spreading? Without knowing this, it is hard to comment one way or another on whether spreading is frequent or an isolated event. It would likely be more appropriate though to provide an analysis of the autocorrelation between hypermethylation events and distance between CpG sites.

It has been shown that DNA methylation spreading can occur after dCas9 3A3L targeting (Fig. 2b, Stepper et al PMID: 27899645), because of this we hypothesised that any off-target dCas9 3A3L mediated effects may show a similar spread of DNA hypermethylation. Although the data from *GOS2* and *C10orf41/ZNF503-AS2* is suggestive of DNA methylation spreading (Supplementary Fig. 4a, b), we agree that since we have not done a thorough analysis of DNA methylation spreading we cannot conclusively state this and so have removed the text.

5) p.4 line 164. I found the statement here overly strong, redundant, and a bit out of place, since it refers to data later in the paper. I would just remove. I think the statement in the discussion in page 10, line 423 is sufficient.

We agree with the reviewer and have removed this sentence.

Reviewer #3 (Remarks to the Author):

The manuscript is improved in this revised version. My comments have been satisfactorily addressed. However, some areas are still a little difficult to understand and can be further improved with copyediting. Below are a few suggestions from just the first page:

In some instances, it would be better to use "inability to engage senescence arrest" rather than "prevented from entering senescence" (abstract, lines 42-43).

The last sentence of the abstract is confusing. A modification more appropriate would be, "This work demonstrates that hit-and-run epigenetic events can prevent senescence arrest, which may facilitate the early stages of tumor initiation" or something similar.

The third “Highlight” bullet point suggests hyper-proliferation is driven by p16 when that is clearly the opposite of the actual truth.

We thank the reviewer for their comments and have incorporated them into the main text.

Finally, in the first round of review, Referee 2 pointed out in their minor comment, point 3 that the fold-change shown from the RNA-seq data in Fig. 2h, did not seem to match the significance shown on the graph. In our answer, we explained this was because the significance was calculated using DESeq2, a differential expression analysis package from R, but the graph was plotted based on the RNA-seq pipeline in SeqMonk. To further investigate why the SeqMonk analysis looked this way, we contacted Simon Andrews, Head of Bioinformatics at the Babraham Institute, and he plotted the whole gene expression levels of RASSF1, CDKN2A, PTEN and HIC1 with confidence intervals applied to them (see below). This explains why genes such as PTEN were not found to be significantly upregulated from DESeq2, even though there difference looks substantial from our RNA-seq graph